# Genome and single-cell RNA-sequencing of the earthworm *Eisenia andrei* identifies cellular mechanisms underlying regeneration

Yong Shao[1,13], Xiao-Bo Wang[2,13], Jin-Jin Zhang[1,3,13], Ming-Li Li[1,3,13], Shou-Song Wu[4], Xi-Yao Ma[1], Xue Wang[5,6], Hui-Fang Zhao[5,6], Yuan Li[7], Helen He Zhu[5,6], David M. Irwin [1,8,9], De-Peng Wang[7], Guo-Jie Zhang [1,10,11,12✉], Jue Ruan[2✉] & Dong-Dong Wu [1,12✉]

The earthworm is particularly fascinating to biologists because of its strong regenerative capacity. However, many aspects of its regeneration in nature remain elusive. Here we report chromosome-level genome, large-scale transcriptome and single-cell RNA-sequencing data during earthworm (*Eisenia andrei*) regeneration. We observe expansion of LINE2 transposable elements and gene families functionally related to regeneration (for example, *EGFR*, epidermal growth factor receptor) particularly for genes exhibiting differential expression during earthworm regeneration. Temporal gene expression trajectories identify transcriptional regulatory factors that are potentially crucial for initiating cell proliferation and differentiation during regeneration. Furthermore, early growth response genes related to regeneration are transcriptionally activated in both the earthworm and planarian. Meanwhile, single-cell RNA-sequencing provides insight into the regenerative process at a cellular level and finds that the largest proportion of cells present during regeneration are stem cells.

[1] State Key Laboratory of Genetic Resources and Evolution, Kunming Institute of Zoology, Chinese Academy of Sciences, Kunming, Yunnan, China. [2] Agricultural Genomics Institute, Chinese Academy of Agricultural Sciences, Shenzhen 518120, China. [3] Kunming College of Life Science, University of the Chinese Academy of Sciences, Kunming, Yunnan 650223, China. [4] Guanglong Earthworm Breeding Institute, Lingshan, Guangxi, China. [5] State Key Laboratory of Oncogenes and Related Genes, Renji-Med X Stem Cell Research Center, Ren Ji Hospital, School of Medicine, Shanghai Jiao Tong University, Shanghai 200032, China. [6] School of Biomedical Engineering & Med-X Research Institute, Shanghai Jiao Tong University, Shanghai 200030, China. [7] Nextomics Biosciences Institute, 430000 Wuhan, Hubei, China. [8] Department of Laboratory Medicine and Pathobiology, University of Toronto, Toronto, ON M5S 1A8, Canada. [9] Banting and Best Diabetes Centre, University of Toronto, Toronto, ON M5G 2C4, Canada. [10] Section for Ecology and Evolution, Department of Biology, University of Copenhagen, Copenhagen DK-2100, Denmark. [11] China National Genebank, BGI-Shenzhen, Shenzhen 518083, China. [12] Center for Excellence in Animal Evolution and Genetics, Chinese Academy of Sciences, Kunming, China. [13] These authors contributed equally: Yong Shao, Xiao-Bo Wang, Jin-Jin Zhang, Ming-Li Li. ✉email: guojie.zhang@bio.ku.dk; ruanjue@caas.cn; wudongdong@mail.kiz.ac.cn

Regeneration is one of the most complex and intriguing biological processes that can occur throughout the lifetime of some organisms. However, the regenerative capacity of many animals is extremely limited; in contrast to differentiated tissues and organs, only fetal tissues can be recreated without fibrosis[1]. However, some organisms still retain strong regenerative abilities. For example, the zebrafish, salamander, axolotl, and gecko are regenerative vertebrates that can scarlessly heal wounds and regenerate lost organs and appendages such as fins, heart, jaws, limbs, tails, gills and lenses[2–6]. In addition to vertebrates, several invertebrates such as planarian and hydras possess even stronger regenerative ability and can regenerate almost an entirely new organism because of the abundance of somatic stem cells (neoblasts) in their bodies[7,8]. Therefore, these animals are often viewed as important models in stem cell biology and regenerative medicine.

Charles Darwin performed a large amount of work emphasizing the importance of the earthworm on soil formation and ecosystem development[9]. Earthworms influence the physical characteristics of the soil as they dig burrows, deposit casts on the soil surface and within it and overturn dead organic matter[10]. Because of the remarkable traits of the earthworm in evolutionary biology, such as flightless and legless locomotion, unfathomed diversity, evolutionary conservation, and ecotoxicology, the earthworm may be elevated from the status of a soil sentinel to that elusive entity, an ecologically relevant genetic model organism[9]. In addition, the earthworm is capable of regeneration and has significant benefits compared to planarian and hydras for exploring regenerative mechanisms, which include the following features[11,12]: (1) Complex phenotypic structures, such as an advanced central nervous system with memory function, a closed blood-vascular system, a coelom, and specialized body segmentation. Wound repair involves multiple tissues and is a complex regenerative process; (2) A relatively short regenerative cycle. *Eisenia andrei* and *Perionyx excavatus* can completely regenerate an amputated tail within 35 and 25 days post-amputation, respectively, and *P. excavatus* can complete anterior regeneration with restructuring of reproductive organs (i.e., testis, ovary, seminal vesicle, and clitellum) within 2 weeks of amputation[13]; (3) Bidirectional regeneration capacity. Apart from regenerating an amputated tail, the earthworm can regenerate an amputated anterior portion consisting of the brain, heart and clitellum. Taken together, this collection of phenotypes suggests that the earthworm could serve as an excellent animal model to deeply explore regenerative mechanisms and provide a valuable resource for regenerative medicine.

In Annelida, only three whole genomes, a marine polychaete (*Capitella teleta*), a freshwater leech (*Helobdella robusta*) and *Eisenia fetida*, have been sequenced[14–16]. To date, our knowledge and understanding of regeneration in earthworms has been limited by this lack of high-quality genomes, which have severely hindered our exploration of the genetic mechanisms underlying regeneration in earthworms. In this study, we utilize the long-read Pacific Bioscience (PacBio) platform to sequence a high-quality *E. andrei* genome and transcriptomes from different regenerative stages to identify the genetic basis of earthworm regeneration. In addition, we use single-cell RNA-sequencing from regenerative earthworm cells to identify markers and differentiated cell categories and define cell differentiation trajectories. In summary, we utilize multiple omics methods with a combined view of genetics and cytology to explore the mechanisms of a complex trait, regeneration, in earthworms.

## Results

### Earthworm genome assembly by single molecule sequencing.
We sequenced the genome of the earthworm *E. andrei* (Fig. 1a)

based on 14.4 million long reads (~80×) produced by the PacBio RS platform. The genome was assembled with several assembly algorithms, and the final assembly version was selected based on continuity and completeness (Supplementary Table 1). The genome size of the final assembly was approximately 1.3 Gb, which was close to the estimated size of 1.28 Gb from k-mer estimation and ~1.3 Gb from flow cytometry (Supplementary Figs. 1 and 2). The assembly exhibited a much better continuity, with a contig N50 size of approximately 740 kb, than the genomes of several other invertebrates with strong regenerative capacity, such as *Macrostomum lignano* (contig N50 = 64 Kb) and *Apostichopus japonicus* (contig N50 = 192 Kb)[7,17] (Supplementary Table 2). We additionally generated ~24×(34.7 Gb) PE150 Illumina-based short reads to correct the sequence errors found at 1% of the contig bases and improved the single-base accuracy of the genome to more than 99.97%. By mapping the short reads to the genome, we estimated that the earthworm genome has a high heterozygosity rate of 1.5 heterozygous sites per 100 base pairs (Supplementary Fig. 3). We further constructed Hi-C[18] libraries to anchor and orient the contigs into superscaffolds. Based on the 379 million paired-end reads covering the genome at ~100×, we anchored and oriented 2970 contigs (1129 Mb, ~85%) into 11 long pseudomolecules (N50 = 111 Mb) through a hierarchical clustering strategy (Fig. 1b–d).

To assess the completeness of our genome assembly, we aligned the short reads and the transcriptome unigenes to the genome and found that over 98.2% of the short reads and ~94.5% of the de novo transcriptome unigenes could be mapped to the assembly, demonstrating the high completeness of the assembly (Supplementary Table 3). We also tested for the presence of 978 conserved BUSCO genes and found that 92.1% of the BUSCO orthologs were completely captured in the assembly (Supplementary Table 4). These results indicate the high integrity and accuracy of our assembled genome. Genome annotation was performed by a series of methods, including de novo, homology-based and transcriptome-based prediction. The nonredundant reference gene set identified 31,817 protein-coding genes (Supplementary Tables 5–7).

### Phenotypic and transcriptomic changes during regeneration.
Some studies have documented transcriptomic and some phenotypic changes of posterior regeneration in the earthworms[15,19], but very few researches are focused on the anterior regeneration[13], particularly for *E. andrei*. Therefore, in the present study, we focused on the anterior regeneration in the earthworm *E. andrei*. Phenotypic observations on multiple time points after the anterior amputation (Fig. 2a–c and Supplementary Figs. 4 and 5), could help us to evaluate whole regenerative processes after anterior amputation (the first four body segments), especially for early regenerative events. Using Ki-67 immunofluorescent labeling, we found that cell proliferation initiated at 24 h post-amputation, and at 48 and 72 h post-amputation the proliferating cells increased rapidly and gradually migrated to the center of cross sections (Fig. 2d and Supplementary Fig. 6). At 5 days post-amputation, the wound healing was fully accomplished and a small blastema appeared in center of the amputation plane (Supplementary Fig. 4). At 6 and 7 days post-amputation, the blastema persistently experienced growth and elongation (Supplementary Fig. 4). Although the newly produced body segments were not observed at 14 days post-amputation, the base of outgrowth has accumulated pigments (Supplementary Fig. 4). At 18 days post-amputation, new body segments arise, and at 28 days post-amputation the obvious body segments take shape in regenerative appendages (Supplementary Fig. 4).

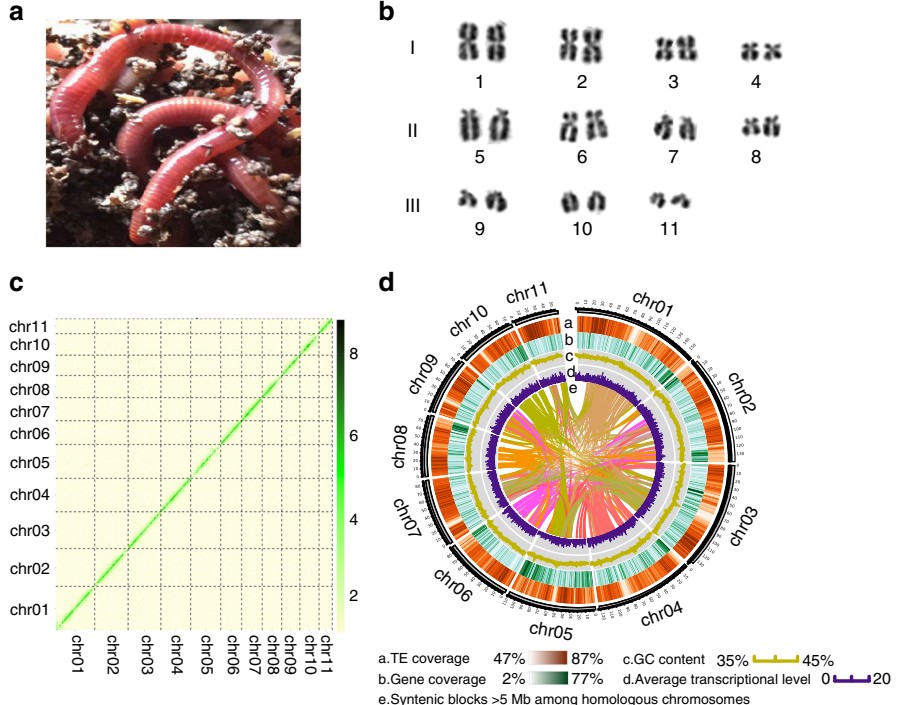

**Fig. 1 Genome assembly of the earthworm. a** A earthworm strain (*E. andrei*) sequenced by the PacBio platform. **b** Karyotype of the earthworm. $2n = 22$. Chromosomes were classed into three subtypes according to their morphological differences. The karyotype was obtained from metaphase cells in the earthworm clitellum. These analyses were performed by Conservation Genetics CAS Kunming Cell Bank. **c** Hi–C interactions among 11 chromosomes. Strong interactions were indicated in dark green and weak interactions were indicated in light green. **d** Circos diagram depicting the characteristics of the earthworm genome. The tracks from outer to inner circles indicated the following: chromosomes, TE coverage, gene coverage, GC contents, gene expression and syntenic block >5 Mb among homologous chromosomes, respectively.

To understand the genetic regulatory mechanisms underlying the early regenerative process in earthworms, we further sequenced the transcriptome of head, the first four body segments containing the central nervous system, during the regeneration process at 6 time points (0, 6, 12, 24, 48 and 72 h after cutting, with 5 biological replicates for each time point) (Fig. 2a). Principal component analysis of the gene expression profiles clearly split the 0 hour time point from the remaining regeneration stage transcriptomes, indicating a high level of gene activity remodeling initiated by the regeneration process (Supplementary Fig. 7). Differentially expressed genes (DEGs) were identified for each regeneration stage compared to the control stage (Fig. 2e; fold change>2 and false discovery rate (FDR) < 0.05). In total, 6,048 DEGs that changed their expression at one or more regeneration time points were identified, and these genes demonstrated a temporal order in their expression profiles (Supplementary Fig. 8). Gene enrichment analysis found that many biological processes important for development were commonly upregulated across all regeneration stages, including gene transcription (GO:0006351), Wnt signaling pathway (GO:0016055), cell surface receptor signaling pathway (GO:0007166), multicellular organism development (GO:0007275), and anatomical structure development (GO:0048856) (Supplementary Data 1). These results indicate that regeneration, as a very complex process, involves multiple genes and pathways. Next, we integrated genomic and transcriptomic analyses to reveal the molecular mechanisms underlying regeneration.

**Expansion of LINE2 transposable elements.** Transposable elements (TEs) make up a large fraction of the genome and play important roles in genome function and evolution[20,21]. In the earthworm, TEs comprise ~56.72% of the genome, posing a challenge for genome assembly (Supplementary Table 8). Among them, DNA transposons and long interspersed nuclear elements (LINEs) comprise the majority of the repeats, spanning 349.6 Mb of the genome (Fig. 3a and Supplementary Table 8). Of particular note, LINE2 has undergone significant expansion (7.49%) in the earthworm compared to other representative metazoan species (2.52% in *C. teleta*, 3.90% in *H. robusta*, 0.00% in *M. lignano*, and 0.84% in *A. japonicus*), and the closely related species *E. fetida* also harbors a high LINE2 proportion (~4.10%) compared to other un-earthworm species, although a low genome assembly quality may underestimate this possibility (Fig. 3b, Supplementary Figs. 9 and 10, and Supplementary Data 2). The number of substitutions to repeat consensus[4], which is an estimate of the relative age of the LINE2, implied that the earthworm LINE2 has undergone a recent and apparent burst of expansion with a peak at 25~30 Mya (Fig. 3c), which is much more recent than its divergence time (309 Mya) from *H. robusta* (Supplementary Fig. 11).

Approximately 43.54% of the LINE2 elements in the earthworm genome are located in intron regions, and 6.66% are located within the 5-kb flanking regions of genes (Fig. 3d). This suggests that the function of LINE2 is potentially involved in regulatory roles. To test it, we performed further analyses by integrating transcriptomes described above. We discovered that the proportion of DEGs (described above) harboring LINE2 elements, was significantly higher than that of non-DEGs (background genes) harboring LINE2 elements (Fig. 3e, $P = 7.641$E-07, $\chi^2$ test). Further, 44 and 119 significantly differentially expressed LINE2 elements (DEL2s, FDR < 0.05), located in 5k 5'-flanking and 5k 3'-flanking of coding genes, respectively, were identified, which potentially were activated during regeneration

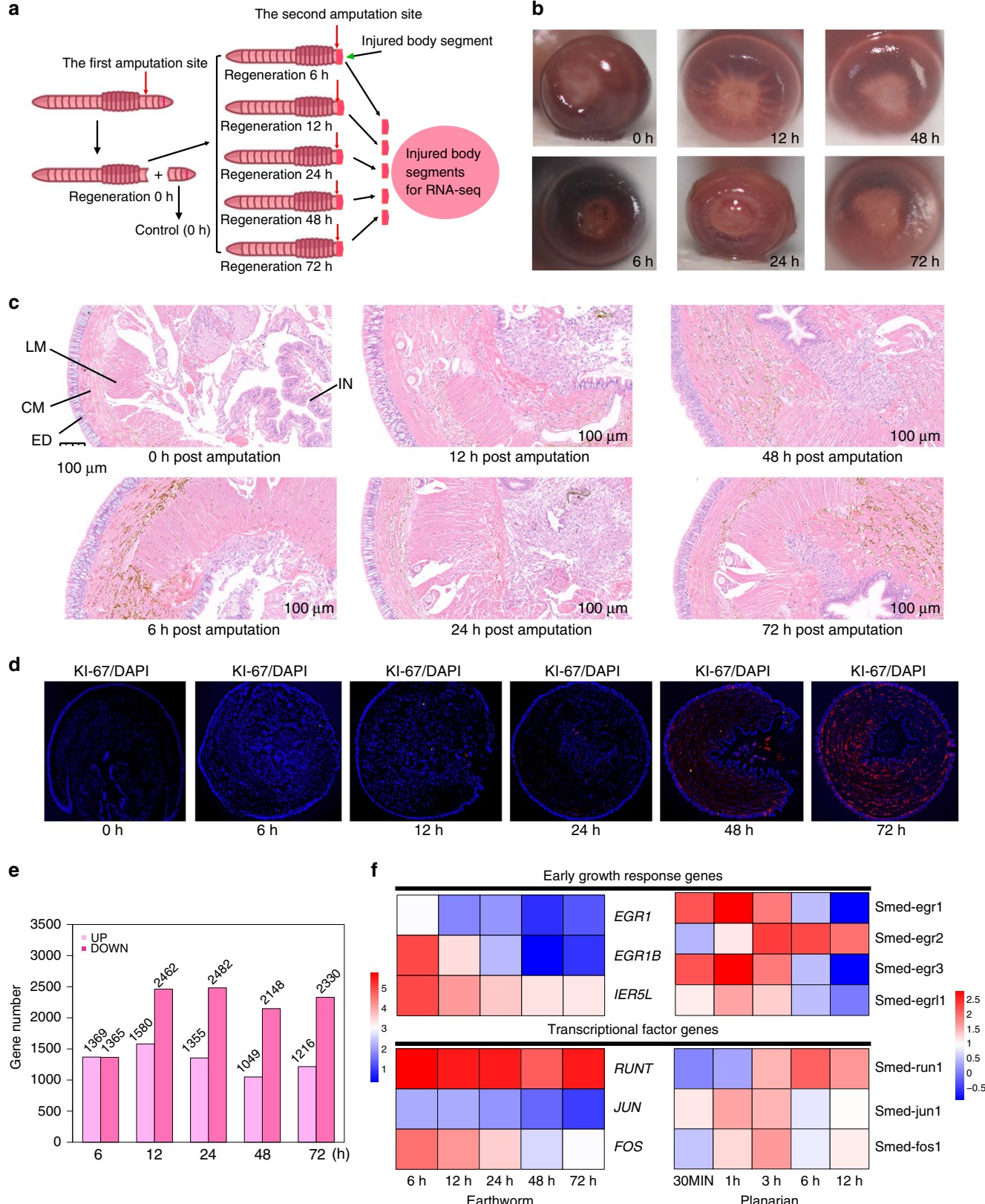

process because of their increasing expression trends (Fig. 3f and Supplementary Fig. 12), especially for DEL2s in 5k 5′-flanking ($P < 0.05$, Mann-Whitney $U$ test). Among these DEL2s within the 5-kb flanking regions of coding genes, we found 19 DEL2s were transcriptionally activated with significantly increased expression during the regenerative process and their neighboring genes also demonstrated similar increasing expression trends (Fig. 3g, FDR < 0.05, Benjamini-Hochberg FDR). The neighboring genes of 19 DEL2s, such as *EGR1*, *FOSL*, *BMP10*, *HUNB* and *MMP17*, are frequently reported to participate in regeneration[22–24]. For example, *EGR1* functions as a pioneer factor to directly regulate early wound-induced genes in acoels[22]. Our analyses suggest

**Fig. 2 Phenotypic and transcriptomic analyses during regeneration. a** A cartoon of time-dependent amputation and regeneration transcriptome sequencing in earthworm. **b** Snapshots of cross sections at six time points after post-amputation. **c** HE staining of cross sections at six time points post-amputation. The different structure layers were labeled. ED: epidermis, CM: circular muscle, LM: longitudinal muscle, IN: intestine. A representative scale bar (100 μm) was showed. The HE staining experiments were independently repeated (three times) and obtained similar results. **d** Detection of cell proliferation at different time points post-amputation using a marker Ki-67 by Immunofluorescent double staining. The red fluorescence represented signals and the blue fluorescence represented cell nucleus. Similar results were reproduced in two independentbiological experiments. **e**, Number comparisons of DEGs at different regeneration time points, compared to regeneration 0 h (controls). Time points included 0, 6, 12, 24, 48 and 72 h. **f** Co-activation of gene expression in early regeneration processes between earthworm and planarian. The early growth response genes and transcriptional factor genes respectively were compared for two species (earthworm and planarian). The planarian gene 1080 expression changes were obtained from a previous study[37]. The dark red represented genes with higher expression levels, and the dark blue showed genes with lower expression levels.

that partial LINE2 elements in earthworms might regulate the expression of neighboring genes by coopting them into regeneration-regulatory networks. However, we acknowledge that further experiments are needed to elucidate how LINE2 elements regulate gene expression during earthworm regeneration. Overall, our study suggests that LINE2 elements in earthworms may play important roles in early regenerative processes.

**Evolution of gene families in the earthworm genome**. Expansion or contraction of gene families is associated with the evolution of specific phenotypes and physiological functions. In the present study, we identified 26,926 gene families from 12 invertebrates (Supplementary Fig. 13). 4,877 gene families were shared by five species (*E. andrei*, *H. robusta*, *C. teleta*, *Crassostrea gigas*, and *Lottia gigantea*) (Supplementary Fig. 14), while 1165 gene families were unique to earthworms (Supplementary Fig. 15 and Supplementary Table 9). In line with a previous study[16], which identified extensive gene duplications functioning as regulating early development in the *E. fetida* genome, we also found abundant expanded gene families in the earthworms (i.e., +2776 in *E. andrei* and +3537 in *E. fetida*) (Fig. 4a and Supplementary Fig. 16). We further estimated the time of these duplication events by using $K_S$ distributions, where $K_S$ is the synonymous distance or defined number of synonymous substitutions per synonymous site[25]. $K_S$ distributions of duplication events in the *E. andrei* genome were obviously larger than the $K_S$ distribution of one-to-one orthologs between *E. andrei and E. fetida*, which implied that these gene duplications occurred before the divergence of *E. andrei* and *E. fetida* (Fig. 4b). Furthermore, these expanded gene families were mainly enriched in GO terms including cell-cell signaling (GO:0007267, $P = 2.38E-02$), Wnt signaling pathway (GO:0016055, $P = 2.32E-02$), cell surface receptor signaling pathway (GO:0007166, $P = 6.05E-03$), regulation of cell communication (GO:0010646, $P = 3.91E-06$), development process (GO:0032502, $P = 4.19E-05$), ion transport (GO:0006811, $P = 5.47E-03$), organelle organization (GO:0006996, $P = 3.56E-02$), and regulation of cellular biosynthetic process (GO:0031326, $P = 7.98E-05$) (Supplementary Data 3). We speculated members of these expanded gene families in *E. andrei* may potentially participate in special phenotypic evolution of the earthworm, such as regeneration. Similarly, a previous study using expressed sequence tags also found that biological processes such as cell-cell communication and biosynthesis could occur during the regenerative stages in *P. excavates*, another earthworm[13]. Of particular interest, the Wnt signaling pathway, a canonical regeneration pathway controlling anteroposterior polarity during planarian regeneration[26–29] and regulating progenitor cell fate and proliferation during regeneration of zebrafish fins and deer antlers[30,31], have displayed a substantial expansion in the earthworm. For example, the genes *APC* and *DVL3* showed expansions in the Wnt signaling pathway and exhibited increasing trends in expression during regeneration (Supplementary Fig. 17).

Among 186 significantly expanded gene families in the earthworm branch (Viterbi *P*-value ≤0.05), 35 gene families harbor over 10% of their family members displaying significant expression changes during regeneration (Fig. 4c and Supplementary Fig. 18). Furthermore, we performed a randomization test and found five gene families standing out as showing significantly higher proportion of differentially expressed genes ($P < 0.05$, $\chi^2$ test), including *ZNFX1, EGFR, NNP, HELZ2* and *SACS*. For example, *ZNFX1* is activated in both newt and axolotl incompetent iris regeneration[32], and 9 of 11 copies of *ZNFX1* exhibit significant expression changes in earthworm during regeneration processes ($P = 0.0105$, $\chi^2$ test, and Supplementary Fig. 19). Gene, *EGFR*, encodes an epidermal growth factor receptor, which is a transmembrane receptor with tyrosine kinase activity that can regulate cell proliferation and differentiation[33]. In planarians, silencing of *EGFR-1* and *EGFR-3* can result in abnormal morphogenesis and disorganized developmental structures during regeneration[33]. *EGFR* experienced a significant expansion with a significantly increased copy number in the earthworm (12 copies) relative to other species, which have 0~2 copies (Fig. 4d, and $P = 0.0114$, $\chi^2$ test). Eight of the 12 members showed differential expression levels during regeneration and real-time Quantitative PCR further validated expression trends of these duplications in regenerative process of the earthworm (Fig. 4e, Supplementary Fig. 20, and Supplementary Table 10). Although these gene families have diverse roles during development across life cycle, the members of them were significantly differentially expressed during the earthworm regeneration processes and their duplications might potentially play a role in the evolution of regeneration in *E. andrei*.

**Temporal gene regulation patterns in regeneration**. To understand the large-scale gene interactions involved in regeneration, we conducted a weighted gene coexpression network analysis (WGCNA)[34]. This quantitative network-based approach has proven to be a powerful tool for elucidating cell type, anatomic and convergent gene networks across species[35]. Here, we identified 19 gene coexpression modules in response to temporal changes during the regeneration process (Fig. 5a and Supplementary Figs. 21–24). These modules represent genes that share highly similar expression patterns during regeneration (Fig. 5a).

Of these 19 modules, five modules (tan, brown, lightcyan, grey60, and cyan) were dominated by genes showing upregulation at the early stage of the regeneration process (6 h) (Supplementary Figs. 23, 24). Among them, the expression of the brown module was most significantly correlated with the regeneration stage (6 h) ($r = 0.53$, $P = 0.003$) (Fig. 5b and Supplementary Fig. 23). Genes enriched in this module participate in signal transduction, transcription and translation, implying an increasing level of cell communication and biochemical processes via the synthesis of mRNAs and proteins in response to regeneration (Supplementary Data 4). The list of driver genes in the brown

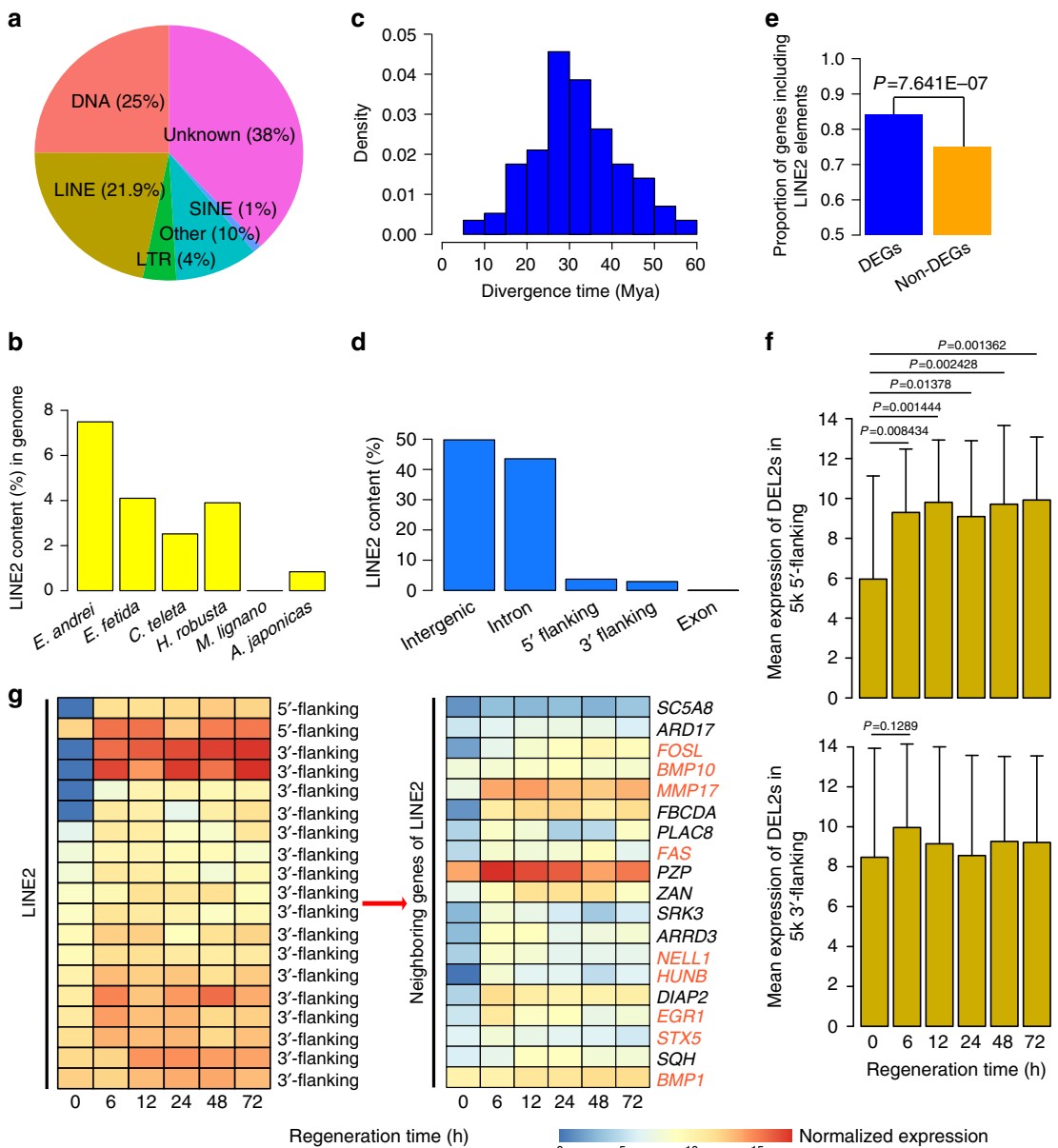

**Fig. 3 LINE2 transposable elements are related to regeneration in earthworm. a** Pie of the major repeat classes in earthworm genome. LINE: long interspersed nuclear elements; SINE: short interspersed nuclear elements. **b** Comparative analyses of LINE2 contents in the genomes across different invertebrates. **c** Divergence time of LINE2 in the earthworm genome. Kimura nucleotide distance of masked regions against their consensus sequences are automatically estimated by RepeatMasker, and the L2 element age was calculated using a mutation rate of $2.7 \times 10^{-9}$ (in *C. elegans*). **d** Distribution trends of LINE2 in the earthworm genome. **e** Proportion of DEGs harboring LINE2 (5065/6048) significantly surpassed the proportion of non-DEGs harboring LINE2 (19,421/25,769) ($P = 7.641E{-}07$, $X^2$ test with Yates' continuity correction) during regenerative process in earthworm. **f** Mean expression values of 44 DEL2s ($n = 44$) in 5k 5'flanking of coding genes and 119 DEL2s ($n = 119$) in 5k 3'flanking of coding genes during regeneration process. Expression value of each DEL2 was normalized by Log$_2$(expression+1) for each time point after post-amputation. The significant levels were calculated by Mann-Whitney $U$-test with continuity correction. The error bars were calculated by using standard deviation (s.d.). **g** Similar expression profile patterns between significantly differentially expressed LINE2 located in 5-kb flanking regions of coding genes and their corresponding neighboring genes showing significant expression changes during regeneration in earthworm. Dark orange represented higher expression levels, and dark blue represented lower expression levels. The representative genes associated with regeneration were highlighted.

module triggered by the regeneration process includes several genes involved in cellular proliferation, differentiation and programmed cell death, such as *FOS* (intramodule membership = 0.9587) and *HUNB* (intramodule membership=0.934) (Fig. 5c, and Supplementary Table 11). Previous studies reveal that *FOS* participates in neoblast maintenance and the wound response program in planarians[36,37] and is a key factor in the cell signaling system activated immediately after cell damage[38].

Two other modules, red and blue, containing genes with increased expression until 12 h of regeneration (Fig. 5d, e, and Supplementary Fig. 24), were also enriched in genes involved in biosynthetic processes and the regulation of cell growth (Supplementary Data 5 and 6). Additionally, the blue module was enriched in genes involved in energy metabolism that are necessary for cell proliferation and growth. However, intriguingly, both of these networks lacked core driver genes

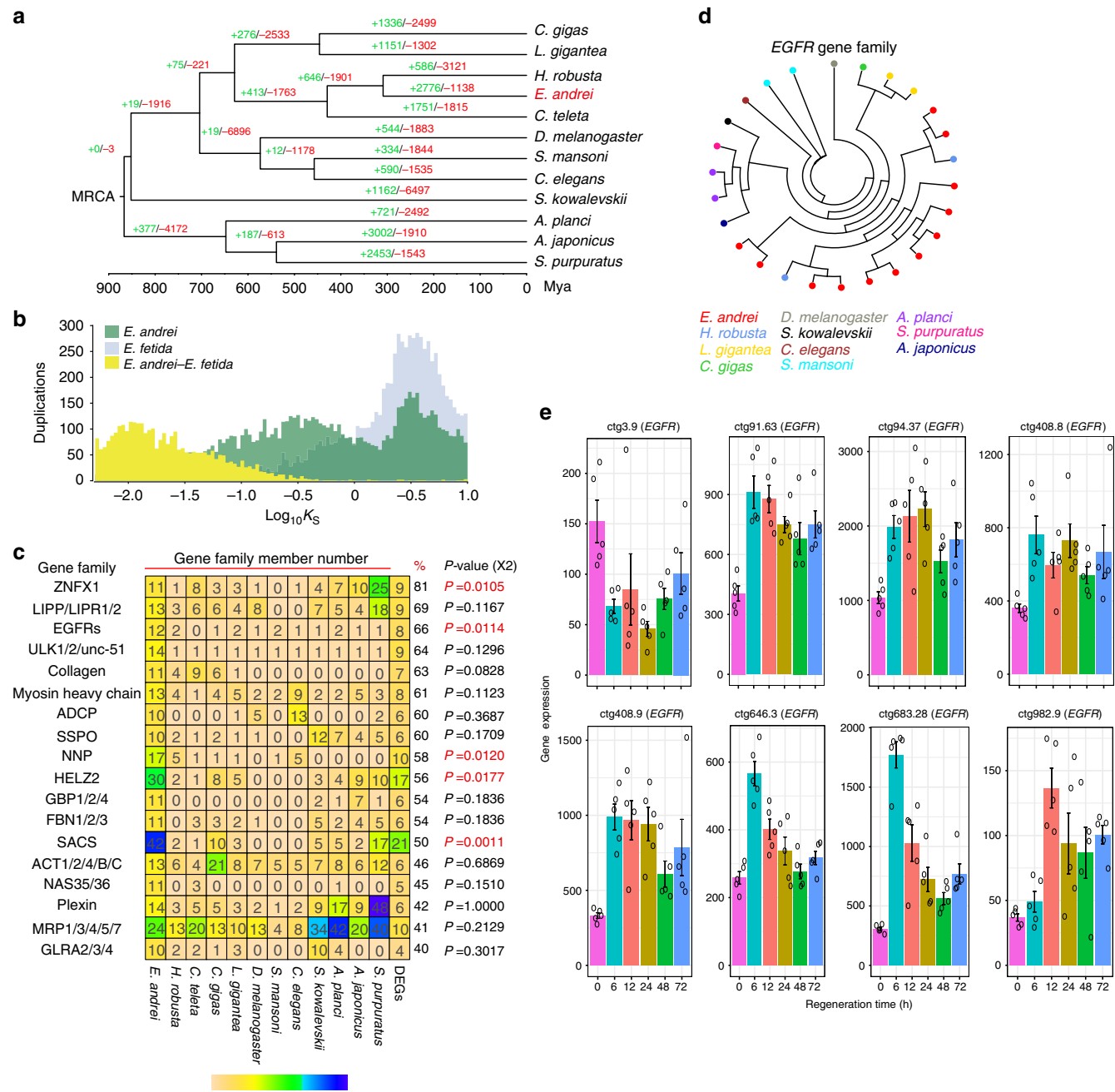

**Fig. 4 Evolution of gene families in the earthworm genome. a** Expansion/constraction of gene families for 12 invertebates. Expanded gene families were shown in green and contracted gene families in red at the whole genome levels. **b** *E. andrei* and *E. fetida* paranome $K_S$ distribution and $K_S$ distribution of one-to-one orthologs of *E. andrei* and *E. fetida*. We constructed and visualized the $K_S$ distribution of paralogs and orthologs using 'ksd' with default parameters and 'viz' command in 'wgd' tools, respectively. **c** Gene families possessing higher proportions of time-dependently DEGs in regeneration. The numbers in the grids were the copy numbers of the gene families for each species. The % column showed the proportion of DEGs in the specific gene family. And the gene families with higher copy numbers were indicated in dark blue, and the gene families with lower copy numbers were indicated in light yellow. For each candidate gene family, a random test was performed by comparing observed numbers of DEGs in regeneration and randomly numbers by using $\chi^2$ test. The significant gene families were highlighted with using red *p*-values. **d** The epidermal growth factor receptor gene family has a significant expansion in the earthworm compared to other invertebrates. The earthworm has 12 copies (red color), and the other included species only 1–2 copies. Epidermal growth factor receptor (*EGFR*). And different species were showed by using diverse colors. **e** Transcriptomic analyses of *EGFRs* during regenerative processes in earthworm. For each regenerative stage of each copy included 5 biological replicates ($n = 5$). The error bars were showed by using standard error of the mean (s.e.m.). And copies were named by contig' orders.

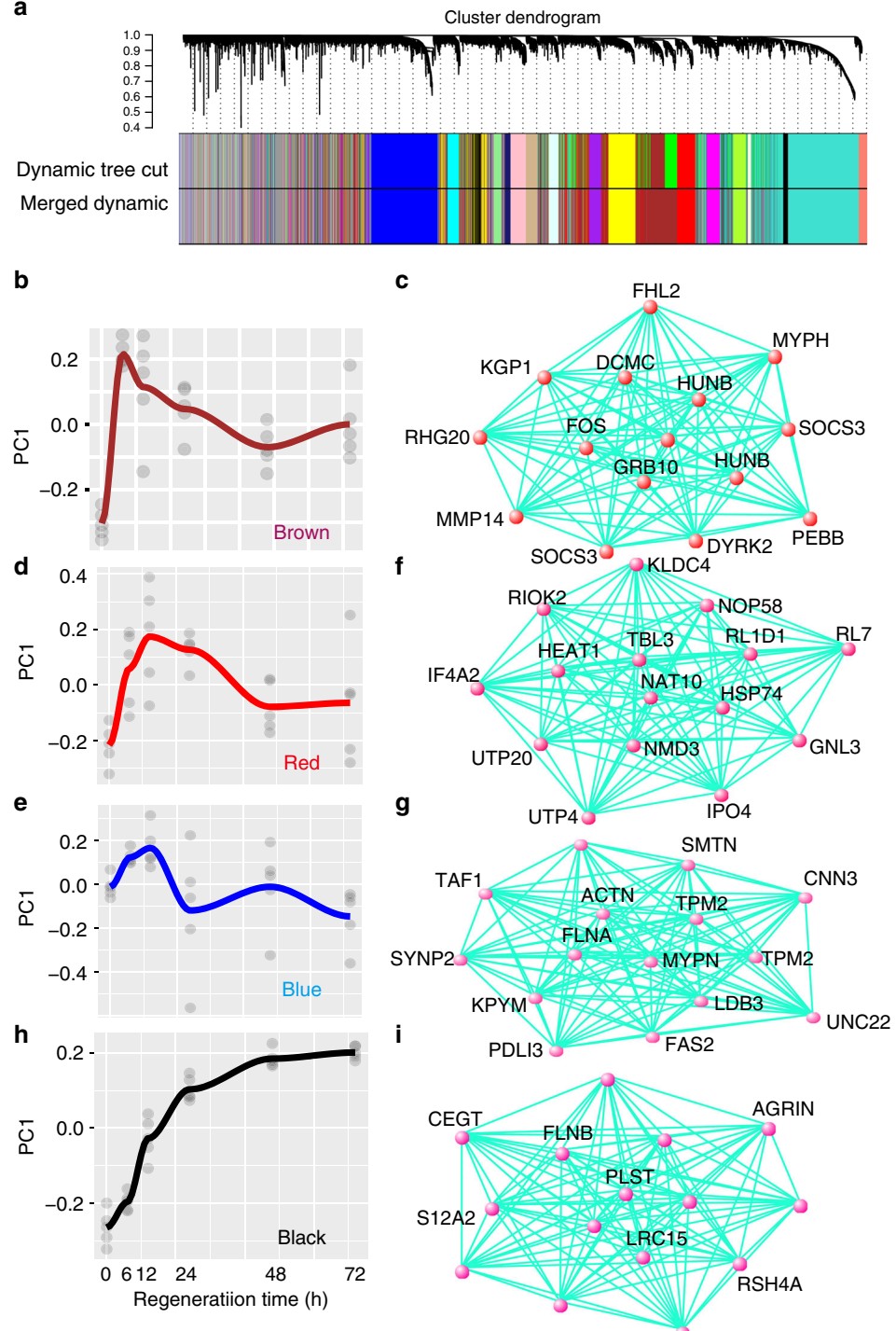

**Fig. 5 Coexpression network analysis in regeneration. a** Network analysis dendrogram showing modules based on the coexpression topological overlap of genes throughout regeneration. Colored bars below give information on module membership. **b** The expression trajectory of brown module throughtout regeneration. **c** Top15 driver genes of brown module. **d** The expression trajectory of red module throughtout regeneration. **e** The expression trajectory of blue module throughtout regeneration. **f** Top15 driver genes of the red module. **g** Top15 driver genes of the blue module. **h** The expression trajectory of black module throughtout regeneration. **i** Top15 driver genes of the black module. For **b**, **d**, **e** and **h** the fit line represented locally weighted scatterplot smoothing. *X*-axis represented regenerative time points and *y*-axis represented the module eigengene. For **c**, **f**, **g** and **i**, top15 drivers were obtained by WGCNA network connection algorithm with the threshold of 0.03, and the drivers without symbols were undefined protein-coding genes. Those drivers with identical symbols were annotated as homologs.

functioning as regeneration regulators (Fig. 5f, g). Therefore, we proposed that the two modules might be vital for regulating the preparation process of the cell proliferation in the early phase of earthworm regeneration.

The black module contains genes that exhibit upregulation within 6 hours after amputation and then gradually increase in expression until 72 h (Fig. 5h, $r = 0.49$, $P = 0006$, and Supplementary Figs. 23, 24). This module presumably has an important

functional role, especially at 48 and 72 h of the early phase of regeneration, because of its sustained and increasing activity. Gene enrichment analysis found that this module was significantly enriched in genes with functions in phosphorylation, cell surface receptor, enzyme activity and ATP binding, all of which are vital for signal transduction (Supplementary Table 12). We uncovered driver genes in the black module, such as *AGRIN*, which had a higher network connectivity (intramodule membership = 0.9276) and is a component of the extracellular matrix, affecting regenerative capacity and development processes in mammals[39,40] (Fig. 5i and Supplementary Table 13). Thus, we propose that the black module genes, with their increasing consistent temporal regulation patterns, may play an important functional role in earthworm regeneration.

**Transcriptional activation of immediate early response genes**. We next sought to discover genetic toolkits that participate in the wound-induced regeneration processes of earthworms and planarians by comparing the temporal transcriptome data from these two species[37]. We found the early growth response genes were transcriptionally induced as a rapid response to injury healing in both species. In earthworms, the expression of the early growth response protein 1 gene (*EGR1*) and the immediate early response gene (*IER5L*) was significantly up-regulated at all regeneration stages, meanwhile the expression level of the early growth response protein 1-B gene (*EGR1B*) was significantly elevated at 6, 12 and 24 hours (Fig. 2f). Similarly, in planarians, we noticed that genes involved in early growth responses, i.e., *EGR1*, *EGR2*, *EGRL1* and *EGR3*, were also transcriptionally activated (Fig. 2f). Importantly, *EGR1* was a shared gene in both of two species during regeneration. *EGR1*, as a member of the immediate early response gene transcription factor family, is implicated in the regulation of multiple cellular processes, such as cell growth, development and stress responses in many tissues, and can control the proliferation and localization of stem cells[41].

Additionally, several important transcription factors, *RUNT*, *JUN* and *FOS*, regulating regeneration processes, were involved in early regeneration in both species (Fig. 2f). For example, the *RUNT* gene, encoding the Runx transcription factor, whose function specifies different cell types during regeneration and promotes heterogeneity in neoblasts near wounds in planarians[37], was significantly upregulated in earthworms throughout the regeneration process and was also upregulated in planarians at 3, 6, and 12 h. Thus, our results suggest the earthworm and planarian potentially utilize a set of similar transcriptional activated immediate early response genes to regulate early regeneration process.

**Single-cell RNA-sequencing reveals cytological mechanisms**. To provide an in-depth understanding of the complex interplay among the molecular and cellular processes underlying earthworm regeneration, we performed single-cell RNA-sequencing using 10X Genomics Chromium platform to examine regenerating heads (the first four segments) at 72 h after cutting. In brief, once the head was amputated, we obtained regenerating segments for cell dissociation and cell sorting. We captured a total of 2080 cells with an average of 493 genes and 1904 transcripts per cell (Supplementary Fig. 25 and Supplementary Table 14). After quality control (QC) filtering, the expression profiles of 2060 single cells were clustered by using Seurat (https://satijalab.org/seurat/). In total, we identified 12 cell clusters using t-SNEs (Fig. 6a).

Next, we elucidated the cell type identity for each cluster by screening marker genes and constructing single-cell trajectories corresponding to a developmental process. Many studies have used *OCT4*, *SOX2* and *NANOG*, called the master regulators of

pluripotency, as markers of pluripotent stem cells (PSCs)[42,43]. The homologs of *SOX2* and *ACTB* (a highly expressed marker in gamma neoblasts in planarian)[44] could be identified in the earthworm genome, and accordingly harbored higher expression levels in cell clusters0/1/3 (Fig. 6a–c, and Supplementary Figs. 26–28). Gene enrichment analyses showed that marker of cluster0/1/3 were significantly over-represented in terms involved in stem cell biology (Fig. 6d, Supplementary Fig. 29). Consistently, a mass of cells (cluster0/1/3) were properly located at the root of the developmental trajectories representing the process of cell differentiation (Fig. 6e). These analyses hint that the cluster0/1/3 may represent putative PSCs at 72 h post-amputation in the earthworm. These three cell clusters shared similar gene expression profiles (Fig. 6b) and made up the largest proportion of captured cells (~45%, Fig. 6f). In situ hybridization of markers *SOX2*[42,43] and *H2B*[44] in early phases of regeneration from 0 to 72 h post-amputation showed that at 6 and 12 h post-amputation, the PSCs arise in the circular muscle layer of the body wall (Fig. 7a–c and Supplementary Fig. 30). And then the PSCs rapidly proliferated and migrated to the longitudinal muscle layer near to the epithelium of intestine (EP) at 24, 48 and 72 h post-amputation, while these PSCs didn't turn up in the epidermis layer (Fig. 7d–f and Supplementary Fig. 30).

Meanwhile, marker genes such as *TPM*[44] and *UNC-89*[45] could define cluster2/4 as muscle cells (Supplementary Fig. 31), and furthermore cluster2/4/5/6/8 were promiscuously located in a leaf of the single-cell trajectories, which suggests that cluster5/6/8 might also be referred to as muscle involved cells (Fig. 6e), representing the second most prevalent cell type we captured. Additionally, the identities of a few neuronal cells consisting of cluster7 were detected by neuronal markers, such as *NF70*, *NBAS* and *AHNAK* (Supplementary Fig. 32) and a clear mapping of clusters including cluster7/9/10/11 was identified in a neuron leaf of the single-cell trajectories (Fig. 6e and Supplementary Fig. 32). Future abundant early single-cell regenerative transcriptomes before and after the blastema formation and functional experiments in the earthworm would validate this possibility.

## Discussion

A mounting number of studies suggest the importance of earthworms in terms of understanding many aspects of biology[9]. In particular, earthworms are of great interest from the perspective of regenerative biology[19,46]. To date, apart from *C. teleta* and *H. robusta*[14], which are annelida, the genome of only one other species, *E. fetida*, from oligochaeta (also known as earthworms) has been sequenced using the next generation genome sequencing strategy, but provided poor assembly quality (contig N50 = 1,852 bp and contig N50 = 967 bp, respectively)[15,16]. Having no high quality genome severely hinders the development of earthworm regeneration biology. In this study, we present a chromosome level genome assembly of the earthworm *E. andrei* with a scaffold N50 = 111 Mbp using a single molecule sequencing (PacBio) integrating Hi-C assembly technology, up to now representing an optimal genome assembly in the phylum annelida. The earthworm *E. andrei* exhibits a high level of regenerative ability at both its anterior and posterior and is easy to culture and handle in laboratory[12,19]. Therefore, it can be potentially regarded as a valuable model to investigate the mechanisms underlying regeneration. We believe that this high-quality genome will supply a useful genetic resource for future research especially in regeneration biology.

Increasing genomes from diverse species indicate that nearly half of genome sequences are derived from TEs, which have played important functional roles in many biological processes[47]. In this study, we propose a potential regulatory role of LINE2 in

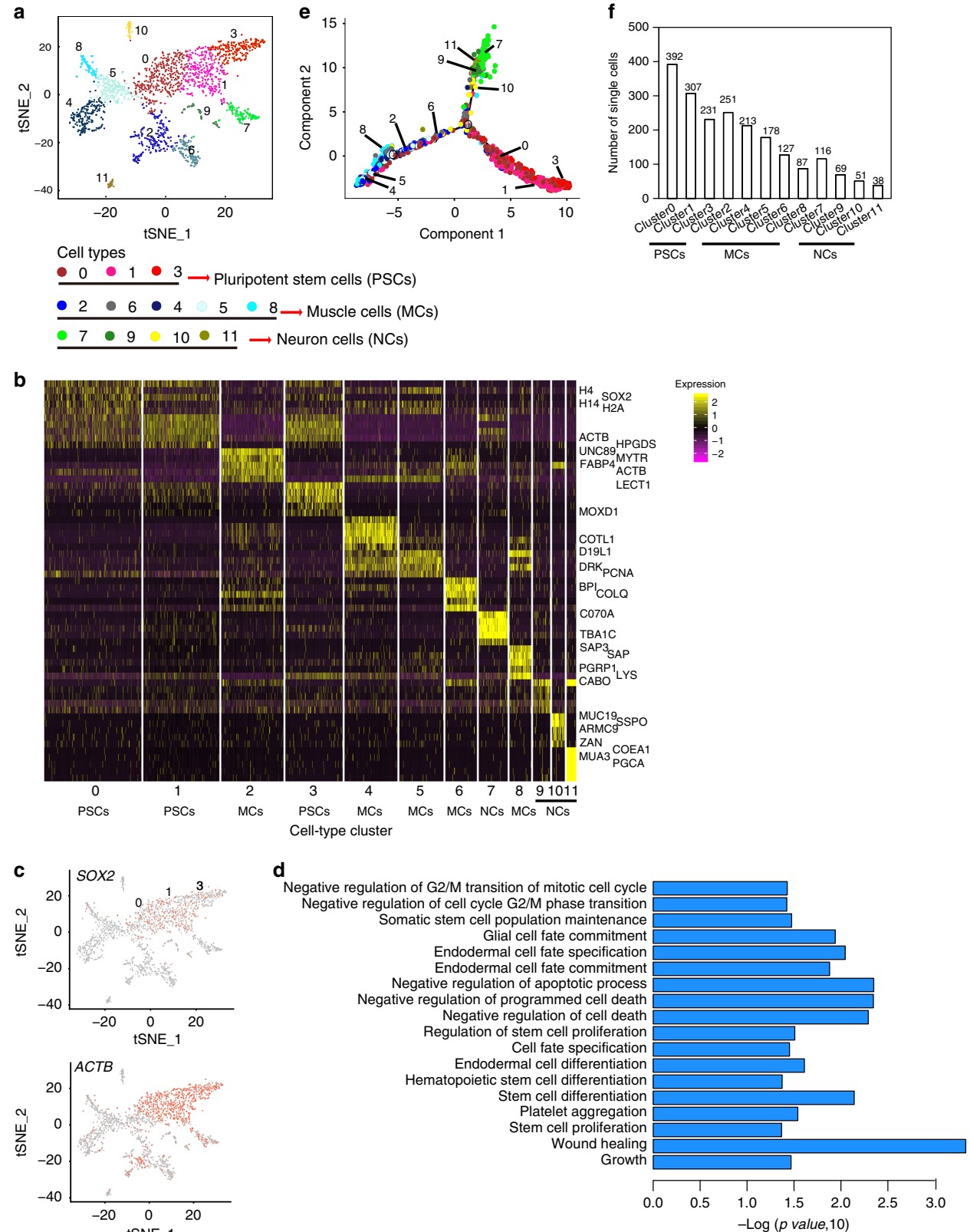

the evolution of the earthworm, possibly in earthworm regeneration. We discover that several LINE2 elements are inserted in the loci of DEGs during early stages of earthworm regeneration. Some specific differentially expressed LINE2 elements in the 5k-flanking sequences of coding genes and their neighboring genes harbored similar increased expression trends during earthworm

regeneration. For example, *EGR1*, a core regulator of wound inducing process in diverse regenerative organisms[48–51], such as in acoel and planarian[22,37], displayed significant differential expression, and harbored differentially expressed LINE2 elements in the earthworm. However, future experiments are required to relate expanded LINE2 with regeneration of the earthworm.

**Fig. 6 Analysis of single-cell RNA-Sequencing during earthworm regeneration. a** Cell clustering plots with single-cell transcriptomic data. And the cell types were coded by Arabic numerals (from cluster 0 to cluster 11). **b** Expression heatmap for different cell clusters. We plotted the top5 highly expressed marker genes for each cluster. Each line represented highly expressed marker genes in specific clusters with the highest expression fold changes, which were compared to all of other clusters. Highly expressed markers were indicated in dark yellow, and the low expressed markers were indicated in dark pink. **c** Characterization of the clusters0/1/3 using two published pluripotent stem cell marker genes in a tSNE plot. Cell types expressing *SOX2* and *ACTB* were colored in red. This analysis illustrated that the single-cell cluster0, 1 and 3 were probably pluripotent stem cells. **d** GO biological process enrichment analyses of highly expressed marker genes in cluster0. GO enrichment analyses were performed by using g:Profiler software (https://biit.cs.ut.ee/). All known genes of statistical domain size using *Homo sapiens* were regarded as background and *p* values (Significance threshold) were adjusted by using Benjamini-Hochberg FDR. **e** Lineage tree reconstruction of cell atlas. A tree-like trajectory in the reduced dimensional space for different cell atlas, and the cell colors were line with **a**. **f** Distribution of cell number in each cluster. PSCs: pluripotent stem cells, MCs: muscle cells, NCs: neuron cells.

Consistent with a previous study[16], a mount of gene duplication events (i.e., many potential expanded gene families) have occurred in the genome of earthworm. These expanded gene families in earthworms were significantly enriched in terms/pathways representing development biology, which potentially reflect partly their roles in regeneration[52,53]. Particularly, some expanded gene families, e.g. *ZNFX1* and *EGFR*, show a higher proportion of their members undergoing significant differential expression during early phases of regeneration in the earthworm. Previous studies indicated that *ZNFX1* (which encodes a NFX1-type zinc finger-containing protein 1) is up-regulated in both newt and axolotl lens regeneration[32]. *EGFR* (coding epidermal growth factor receptor) controls a variety of signals ranging from cell proliferation, differentiation, to morphogenesis during planarian regeneration, and has been proposed to be involved in stem cell maintenance[33,54–56]. Thus, our findings suggest that these gene duplications may potentially utilize increased dosages to regulate gene expression in regenerative process of earthworms.

The evolution of genome sequence only tells a part of the story of regeneration[15,16]; integrating the transcriptional regulation of genes will help to investigate the mechanisms underlying regeneration of earthworms[11,15,57]. For example, the changes of gene expression in a series of long time scale post-amputation (i.e., 15d, 20d and 30d) have been investigated in *E. fetida* regeneration[15]. However, the molecular regulation during early phases of anterior regeneration still remains largely unclear in the earthworm. Considering that the wound healing process is accomplished at 3–5 days post-amputation in *E. anderi*[19], we performed transcriptomic analyses at early phases of wound healing in this earthworm. Immediate early response genes (e.g., *EGR1*) were transcriptionally co-activated in the earthworms and planarians, implying a set of parallel activated mechanisms in early phases of regeneration. Four vital gene co-expression network modules (i.e., brown, blue, red and black) were identified and these show substantial transcriptionally activation during early phases in earthworm regeneration. Functional enrichment of some of the genes expressed in these networks identified signal transduction, biosynthetic processes and the regulation of cell growth, suggesting that these genes may regulate wound healing process in the early phase of the earthworm regeneration.

The epimorphic process of earthworms is thought to occur mainly via dedifferentiation and subsequent redifferentiation of cells, without any contribution from totipotent stem cells (or neoblasts)[19,58,59], and this process commonly involves blastema formation (dedifferentiated cells), which contributes to redifferentiation in regeneration of *Enchytraeus japonensis* and *E. anderi*[19,59]. The histological observations of blastema formation at 1–3 days post-amputation during *E. anderi* tail regeneration showed that at 3 days post-amputation, the blastema cells, which are likely to be pluripotent cells, rapidly proliferated and migrated to coelom[19]. Here, we performed single-cell RNA-sequencing data at 3 days (72 h) anterior post-amputation in the earthworm, and found that the pluripotent stem cells, potentially representing blastema cells, were the largest proportion of cells at this time. Further ISH experiments supported large proportion of PSCs and found that highly enriched PSCs surrounding the EP (central area) of the cross section, which was consistent with formation of blastema at this time[19,59] (Fig. 7 and Supplementary Fig. 30). However, single cell RNA-sequencing data from more different times will undoubtedly help to understand cellular process of regeneration.

Our study identifies some candidate genetic mechanisms underlying regeneration and highlights the earthworm as a promising model for future studies of regenerative biology. In the future, multiple OMICS strategies, interdisciplinary and functional experiments will provide further insight into the regenerative biology of the earthworm.

## Methods

**DNA isolation, PacBio library preparation and sequencing.** One live earthworm (*E. andrei*, originating from Guangxi province in China) was prepared, and its intestinal tract was removed. After washing with saline solution, the earthworm genomic DNA was collected using a Qiagen kit. After assessing the quality of the DNA, we constructed a PacBio library with an insert size of 20 kb and utilized a single molecular RS sequencer to perform long-read sequencing. Hi-C was performed using the following protocol: the adult earthworm tissues were fixed in 1% formaldehyde solution. The nuclear chromatin was obtained from the fixed tissue and digested using *Hin*dIII (New England Biolabs). The overhangs resulting from *Hin*dIII digestion were blunted by bio-14-dCTP (Invitrogen) and the Klenow enzyme (NEB). After dilution and religation using T4 DNA ligase (NEB), the earthworm genomic DNA was extracted and sheared to a size of 350–500 bp with a Bioruptor (Diagenode). The biotin-labeled DNA fragments were enriched by utilizing streptavidin beads (Invitrogen) to further finish library preparation.

**Estimation of earthworm genome size.** The k-mer algorithm was applied to evaluate the earthworm genome size. The 17 k-mer and 34.7 Gb next-generation sequencing reads were utilized in this analyses. Flow cytometry analysis further was used to evaluate the genome size of the earthworm. In brief, after cell suspensions were prepared, we added 500ul PI (C0080, Solarbio) dye working solution [0.85× PBS 9.4 ml, PI (1 mg/ml) 500 μl, DNA free Rnase (10 mg/ml) 50 μl, Triton X-100 10 μl, Sodium Citrate 10 mg; keep away from light] into the prepared earthworm cell suspension, chicken blood cell solution and mixture of earthworm cell and chicken blood cell, and then they were mixed and moved 400ul to flow tubes covered with fresh-keeping films to be tested. The estimation of genome size was performed using BD LSR Fortessa flow cytometer (BD Biosciences, USA). The genome size of chicken (*Gallus gallus* GRCg6a) (1.04 pg) was utilized as a reference control. Flow cytometry analysis was carried out using the laser excitation at 488 nm with minimum 10,000 events (cells) per sample. The mean fluorescence intensity was obtained using FlowJo (v7.1). The DNA content was estimated using the standard formula for genome size (pg) = (Sample fluorescence channel number FL/ Chicken fluorescence channel number FL) × 1.04 pg.

**Long-read de novo assembly of the genome.** We used ~80X PacBio subreads to perform de novo genome assembly by using Wtdbg (v1.2.7) (https://github.com/ruanjue/wtdbg), FALCON[60] (v052016) and Canu[61] (v1.7). Then, the assembled genome was corrected by aligning subreads using the Arrow program (v2.3.2) with the default parameters. Finally, Pilon (v1.22) was used to polish the resulting assembly with ~24X PE150 reads from the Illumina platform. The base accuracy of the assembly was estimated by Illumina reads alignment. The completeness of the assembly was evaluated by BUSCO genes (http://busco.ezlab.org/). Furthermore, the completeness of the assembly was validated by six de novo transcriptomes using Trinity[62] (v2.1.1).

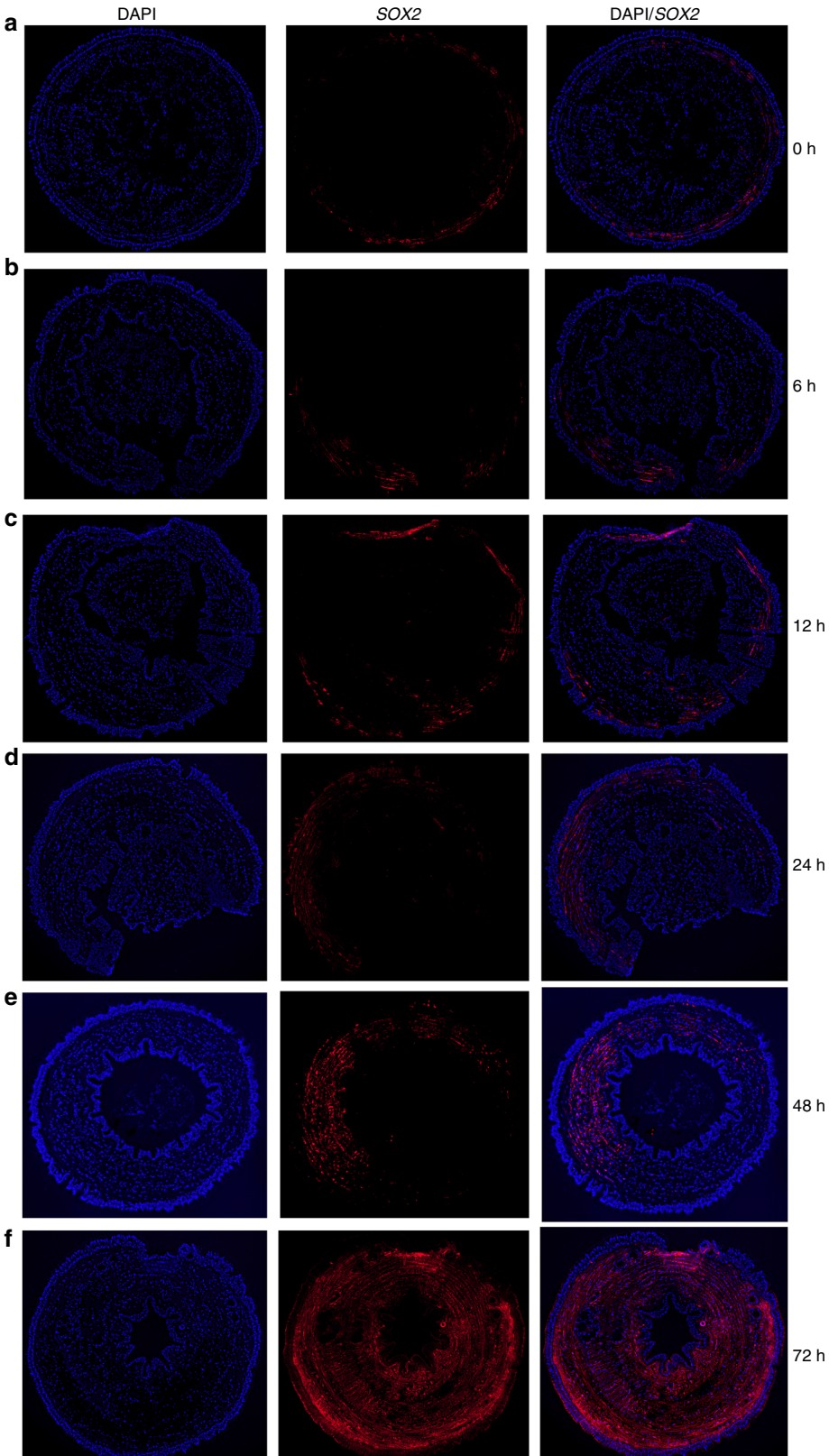

**Fig. 7 In situ hybridization of gene SOX2 in cross sections at 6 different time points post-amputation in the earthworm.** The slice size was 10μm. The 6 time points post-amputation included 0 (**a**), 6 (**b**), 12 (**c**), 24 (**d**), 48 (**e**) and 72 (**f**) hours. The red fluorescence represented positive signals and DAPI (blue fluorescence) was used to stain cell nucleus. Similar results in **a**–**f** could be ensured by three independently biological experiments.

**Genome annotation**. De novo and homology approaches were combined to identify repetitive sequences in the earthworm genome. For the de novo approach, we constructed a de novo repeat library using RepeatModeler (v1.0.8) (http://www.repeatmasker.org/RepeatModeler/) with the default settings. Then, RepeatMasker (v4.0.7) (http://www.repeatmasker.org/) was run on the earthworm genome using the de novo library. RepeatMasker was also run against the RepBase (v20150807) (https://www.girinst.org/repbase/) for homologous repeat identification. The results of repeat annotation using these two approaches were integrated. To annotate the protein-coding genes of the earthworm genome, de novo, homology-based and transcriptome-based prediction methods were combined. Two de novo programs, Augustus[63] (v3.0.3) and SNAP[64] (v2006-07-28), were performed to predict genes in the repeat-masked genome sequences. Long predicted genes processed by PASA[65] (r20140417) were used to train the gene model parameters for the two de novo programs. For the homology-based predictions, protein sequences from *C. teleta* and *H. robusta* (downloaded from the Ensembl database) were aligned to the earthworm genome using tblastn (*e*-value < $10^{-5}$). We used genBlastA[66] (v1.0.138) to cluster the adjacent HSPs (high-scoring pairs) from the same protein alignments, and GeneWise[67] (v2.2.3) was used to identify accurate gene structures. After QC and filtering, reads from all RNA libraries were mapped to the earthworm genome using TopHat2 (v2.0.13) (http://ccb.jhu.edu/software/tophat/), and Cufflinks (v2.1.1) (http://cole-trapnell-lab.github.io/cufflinks/) was subsequently used to predict gene models. All predicted genes from the three approaches were integrated with EVidenceModeler (EVM)[68] (r2012-06-25) to generate high-confidence gene sets. To obtain gene function annotations, KEGG (https://www.genome.jp/kegg/), SwissProt and TrEMBL protein databases (https://www.uniprot.org/) were searched with BLASTP (ncbi-blast-2.2.28+) (*e*-value<$10^{-5}$). The best hits were used to assign homology-based gene functions. Functional classification based on GO categories and InterPro entries was achieved using the InterProScan program (v5.21-60.0) (http://www.ebi.ac.uk/interpro/download/).

**Gene family clusters**. Comparisons among 12 species, including *Caenorhabditis elegans*, *C. gigas*, *C. teleta*, *Drosophila melanogaster*, *H. robusta*, *L. gigantea*, *Schistosoma mansoni*, *Strongylocentrotus purpuratus*, *Saccoglossus kowalevskii*, *A. japonicus*, *Acanthaster planci* and earthworm were conducted to classify gene families. We selected the longest transcript for each gene and eliminated those with premature stop codons, nontriplet length or fewer than 30 amino acids encoded. Subsequently, OrthoMCL[69] (v2.0.9) was used to construct gene families via all-versus-all BLASTP alignments. Changes in gene family size (expansion/contraction) were calculated by the CAFE program (v2.2) (https://hahnlab.sitehost.iu.edu/software.html). To perform phylogenetic analyses, single-copy families were identified, and peptide alignments for each family using MUSCLE (v3.8.31) (http://drive5.com/muscle/downloads.htm) and concatenated to form a supergene for each species. RAxML (v8.2.9) (https://cme.h-its.org/exelixis/web/software/raxml/index.html) with the PROTGAMMAAUTO model and 100 bootstrap replicates was used to build a phylogenetic tree. The peptide alignments were converted to coding sequences, which were subjected to analysis with MCMCtree in the PAML package (v4.8a) (http://abacus.gene.ucl.ac.uk/software/paml.html) to estimate divergence times. Fossil calibration points were obtained from a web-based database—TimeTree (http://www.timetree.org/). If the copy number (gene family) of the detected branch lineage was higher than that of its closely ancestral branch lineage, we regarded this gene family as substantially expanded gene family in this detected branch lineage. The significantly expanded gene families were identified by Viterbi *p*-value ≤ 0.05.

**HE staining**. The fresh tissue was fixed using paraformaldehyde (4%) for 24 h. Afterwards, the tissue was orderly dehydrated using gradient alcohol, and the wax-impregnated tissue was embedded by OCT. And further the tissue was cut into slices with its thickness 4 μm, and the paraffin sections were dewaxed and further washed by distilled water. Lastly, the nucleus and cytoplasm were stained by hematoxylin and eosin, respectively.

**Cell proliferation experiments**. The first four body segments of earthworms were amputated. And at 0, 6, 12, 24, 48 and 72 h post-amputation the injury segments again were amputated for embedding and slicing with a thickness 10 μm, respectively. The Ki-67 was utilized to detect cell proliferation with an anti-ki67 (ab15580) (dilution ratio: 1:200) and a secondary antibody Alexa Fluor 555 donkey anti-rabbit IgG (A31572) (dilution ratio: 1:500). Meanwhile, DAPI was used for staining cell nucleus. The sections were observed using an OLYMPUS TH4-200 inverted fluorescence microscope.

**Transcriptome analysis**. Total RNA was extracted from earthworms at different regeneration time points, including 0, 6, 12, 24, 48 and 72 hours post-amputation (Note that each time point included 5 biological replicates per time point and the wound segment of each individual served as a biological replicate) using TRIzol reagent (Invitrogen Corp., Carlsbad, CA). RNA purifications were performed using the RNeasy Mini Kit (Qiagen, Chatsworth, CA). Sequencing libraries were generated using the NEBNext Ultra RNA Library Prep Kit for Illumina (NEB, USA) following the manufacturer's recommendations. The libraries were sequenced on

an Illumina HiSeq 4000 platform, and 150 bp paired-end reads were generated. After QC, we used TopHat2 to map the clean reads to the assembly reference genome using default parameters. Cufflinks was then applied to assemble the transcripts, with cuffquant and cuffnorm programs in cufflinks used to quantify and normalize transcript/gene expression abundances. DESeq2[70] (v1.26.0) was utilized to construct an expression profile principal component analysis (PCA) to evaluate data quality using the quantified gene expression profile. Cuffdiff in Cufflinks (v2.1.1) was used to detect DEGs between the control (regeneration 0 h) and the case (regeneration 6, 12, 24, 48 and 72 h) samples using a Poisson dispersion model with a FDR ≤ 0.05. The DEGs with expressed changes at one or more regeneration time points were utilized to cluster expression profiles and produce a regeneration spatial-temporal order using the gplots library in R (v3.5.1) (https://www.r-project.org/). If the locus of one gene in DEGs and non-DEGs could overlap at least one LINE/L2 elements, such gene could be regarded as the gene including LINE2 elements.

**Differentially expressed LINE2 elements in coding genes flanking during regeneration process**. We divided the gtf annotation of LINE2 located in 5k flanking of the gene locus into two gtf files including 5k 5′-flanking and 5k 3′-flanking. Then, we respectively mapped our RNA-Seq at different time points (0, 6, 12, 24, 48 and 72 h) after post-amputation to the reference genome according to the two annotations based on the bowtie2 program in Tophat2 (v2.0.13). The expression abundance of each LINE2 was quantified by the cuffquant program in Cufflinks (v2.1.1), and the cuffdiff program in Cufflinks (v2.1.1) was utilized to detect differentially expressed LINE2 (FDR < 0.05) between 0 h and other time points (6, 12, 24, 48 and 72 h) post-amputation. Thus, we screened significantly differentially expressed LINE2 elements in 5′-flanking and 3′-flanking of coding genes, respectively.

**Quantitative real-time PCR**. Earthworms were cleaned using PBS or ddH$_2$O. And we used tweezers to drag earthworms to achieve natural extension and then quickly amputated the first four body segments. The amputated earthworms were placed into soil with fertilizers and cultivated for 0, 6, 12, 24, 48 and 72 h, and then again amputated the injury segments to isolate total RNA by using TRIzol reagent (Invitrogen, 15596-026) and RNeasy® Mini kit (50) (QIAGEN, 74104). The first-strand cDNA was synthesized with 1 μg total RNA using a HiScript® III RT SuperMix for qPCR (+gDNA wiper) kit (Vazyme, R323-01). Quantitative real-time PCR was performed using ChamQ$^{TM}$ Universal SYBR qPCR Master Mix (Vazyme, Q711-03). 5 biological replicates for each time point were guided. The comparative cycle threshold (Ct) method was applied to quantify the expression levels by $2^{(-\Delta\Delta Ct)}$ method. The β-actin was served as a reference gene to normalize the relative mRNA expression levels.

**Identification of coexpression networks in early regenerative processes**. Analysis was carried out in R on a 64-bit LINUX platform with 65.7 GB memory. Modules/or networks were constructed using WGCNA[34] (v1.67). Modules were defined as branches of the hierarchical cluster tree using the dynamic tree cut method. For each module, the expression patterns were summarized by the module eigengene (ME), defined as the right singular vector of the standardized expression patterns. MEs were also defined as the first principal component calculated using PCA, which can summarize module behavior. Pairs of modules with high ME correlations (*R* > 0.8) were merged. MEs for modules were plotted by using the ggplot2 library in R. These MEs were tested for correlation with phenotypes (regeneration time points) adjusted by a linear regression model. In more detail, a weighted signed network was computed based on a fit to scale-free topology, with a threshold softPower of 10 chosen (as it was the smallest that resulted in a scale-free $R^2$ fit). A topological overlap dendrogram was used to define modules with a minimum module size of 80 genes and the deepSplit parameter set to 2. The connectivity of every gene in every module was assessed by correlation to the MEs, or kMEs. Module membership (MM) was regarded as intramodular connectivity. MM can be combined as a systematic biological method to obtain driver genes in networks, which are highly interconnected nodes within coexpression gene modules. The driver genes were defined by the WGCNA connectivity algorithm. Each module network was viewed by VisANT (v5.0) (http://www.visantnet.org/visantnet.html), which allows users to input an edge file and a node file from a WGCNA module.

**Single-cell RNA-sequencing analysis**. The preparation of the earthworm single-cell samples was performed using the following protocol: (1) 15 earthworms were cleaned and soil was removed using PBS or ddH$_2$O. (2) We used tweezers to drag the earthworms to make its head natural extended and then quickly amputated the first four body segments (the brain is located in body segment 3–4 of the anterior). (3) Amputated earthworms were placed into soil with fertilizer and cultivated at 25 ℃ until to 72 h, and then we obtained the wound healing plane segments from 15 earthworms. (4) The mixed wound healing segments were dissociated by adding Collagenase I (500 μl 1 mg/ml) and then maintained 1.5–2 h under 37 ℃. (5) Cells were pelleted by centrifugation at 3000 rpm in 5 min; the supernatant was removed and cell pellets were washed one time using 1× PBS. We then added 200 μl 0.25% TE and allowed the cells to incubate for 5–10 min and then neutralized using 1 ml

1640/DMEM including serum. (6) Cells were again pelleted at 3000 rpm for 5 min, the supernatant was removed and samples were resuspended in 500 μl PBS. Lastly, cell samples were passed through a cell strainer with an aperture 40 μl. (7) Cells were again pelleted at 3000 rpm for 5 min, supernatant was removed and samples were resuspended in 200 μl PBS. (8) Thus, a mixed pool of the earthworm cells (from 15 earthworms) were counted and analyzed by a Flow Cytometer. The Earthworm Single-cell RNA-Sequencing steps is as follows: Chromium$^{TM}$ Single Cell Solution (the experimental protocol) included the following four steps: Cell quality control. We used Countess® II Automated Cell Counter to count cells and adjusted cell concentration to $1 \times 10^6$/ml. (2) 10× marking cDNA fragments. The gel beads including 10X barcode information were first combined with the mixtures of cells and enzymes, and then they were encased by a droplet of oil with surfactant located in a "double cross" connected microfluidic. When the oil droplets flow into the storage chamber and are collected, the gel beads are dissolved and release primer sequences allowing reverse transcription into cDNA fragments. The cDNA was amplified by PCR. (3) Constructing sequencing library. We utilized Biorupter to fragment the cDNAs into 200~300 bp fragments and add sequencing adaptor P5 and primer R1 to perform PCR to obtain a DNA library. (4) Cluster and sequencing. We used Qubit to qualify the sequencing library, and a high-quality sequencing library was placed onto cBot to perform Bridging PCR amplification to regenerate clusters. We then utilized illumina sequencer to complete the sequencing. We used 10X Genomics Cell Ranger (v2.1.1) to perform QC statistics and mapped the data to the earthworm reference genome. Cell Ranger can quantify single-cell transcriptome by differentiating barcode and UMI markers. We used the Seurat (v2) toolkit (https://satijalab.org/seurat/) to analyze the single-cell RNA-sequencing data and cluster the expression profiles of single cells by using the t-SNE method. Furthermore, Monocle (v1) (http://cole-trapnell-lab.github.io/monocle-release/) was utilized to recover single-cell gene expression kinetics and construct single-cell trajectories.

**In situ hybridization**. The second amputated injury segments at 0, 6, 12, 24, 48 and 72 h post-amputation were obtained from *E. anderi*. The tissue sections (10 μm) were stained with DAPI for double staining and they were observed by FITC using an OLYMPUS TH4-200 inverted fluorescence microscope.

**Gene ontology enrichment analysis**. Gene functional enrichments at three levels (biological process, molecular function and cellular component) were performed using a web-based gene analysis tool, g:Profiler (rev1705) (http://biit.cs.ut.ee/gprofiler/). The *p*-value was adjusted by Benjamini-Hochberg *FDR*.

**Reporting summary**. Further information on research design is available in the Nature Research Reporting Summary linked to this article.

## Data availability
The authors declare that all data supporting the findings of this study are available within the Article and its Supplementary Information files or from the corresponding authors upon reasonable request.

The genome assembly and annotation data have been deposited at the Genome Warehouse in the National Genomics Data Center (http://bigd.big.ac.cn/gwh/) under accession code: GWHACBE00000000. The genome sequencing data have been deposited at the Sequence Read Archive (SRA) database at the National Center for Biotechnology Information (NCBI) under accession code: PRJNA541361. The transcriptome sequencing data have been deposited at the NCBI SRA database under accession code: PRJNA541362. The single-cell transcriptome sequencing data have been demonstrated in the NCBI SRA database under accession code: PRJNA541363.

The source data underlying Figs. 1a, 2a–d, 3f, g, 4e, 5b, d, e, h, 6d and 7c and Supplementary Figs. 17, 19, 20, 24 and 29 are provided as a Source Data file.

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

## Acknowledgements

This work was supported by the Strategic Priority Research Program of the Chinese Academy of Sciences, Grant No XDB13020600, National Natural Science Foundation of China (91731304), and the Animal Branch of the GermplasmBank of Wild Species of Chinese Academy of Science (the Large Research Infrastructure Funding).

## Author contributions

D.D.W., J.R. and G.J.Z. designed and leaded the project. Y.S. analyzed genome and transcriptome, and drafted the paper. X.B.W. analyzed the assembly and annotation of genome data. M.L.L. analyzed single-cell RNA-sequencing data. H.H.Z., X.W. and H.F.Z. performed cell dissociations and cell sorting experiments. S.S.W. sampled and processed the experimental materials. J.J.Z. and X.Y.M. did experiments. Y.L. and D.P.W. finished genome sequencing. D.M.I. revised the manuscript.

## Competing interests

The authors declare no competing interests.
