## [Peer Review File · Nature Communications]

Reviewers' Comments:

Reviewer #1:

Remarks to the Author:

In this submitted manuscript, Shao and co-workers present an impressive genomic and transcriptomic study of an annelid species, the earthworm *Eisenia andrei*. Earthworms, which are key species of the soil ecosystem, display several interesting biological properties, including important regenerative abilities. These worms can indeed regenerate lost body structures, in their posterior body region, as well as in several cases (such as *E. andrei*) in their anterior region. Earthworms, and more generally annelids, are interesting models to study regeneration, notably because they have a much more elaborated body plan (with complex organs and organ systems) than other highly studied non-vertebrate species with high regenerative abilities such as flatworms and cnidarians. A better understanding of annelid regeneration is therefore of high interest for the whole regeneration field. In addition, despite the importance of annelids, there only were until now three published full genome sequences from this group.

Shao et al. combined PacBio long-reads, Illumina short-reads and Hi-C sequencing to generate a high-quality genome assembly of *E. andrei* genome, producing what is, to my knowledge, the first chromosome level assembly of an annelid genome. The authors also performed a bulk-RNA-seq analysis at different time points of *E. andrei* anterior regeneration, identifying a large number of differentially-expressed genes during regeneration. They found that LINE2 transposable elements, which underwent a quite recent expansion in *E. andrei*, are often transcriptionally active during regeneration and might have impact on expression of adjacent genes, an interesting hypothesis for which the authors unfortunately did not show experimental evidence. The authors also studied the evolution of some gene families in *E. fetida*, and provide examples of expansion of some families by gene duplications (EGFR and TCAF families). Finally, the authors performed sc-RNA-seq at one time point after amputation, an analysis from which they drew the conclusion that major cell types of the regenerated region are pluripotent stem cells.

This is clearly an important study with a lot of interesting data, and which clearly provides useful insights for our understanding of regeneration in annelids. There are however problems that should be solved by the authors.

Major concerns :

1. One of major concern is the lack of a clear description of *E. andrei* anterior regeneration. The authors took regenerated region at different time points after amputation, but we do not have any idea of what these regions look like at these different time points. When is wound healing completed ? Are there proliferating cells at these different time points ? When are differentiated cells or structures, such as muscles or neural cells, observed ? Is the brain fully regenerated by 72 hours post-amputation ? I think that these are crucial information to be able to make in depth use and interpretation of the nice transcriptomic data generated by the authors. This information should be provided.
2. My second concern is about the section « Evolution of Gene Families Related to

Regeneration », which I found not very clear and misleading. The authors identified gene families that have been expanded in *E. andrei*, including some belonging to particular pathways such as Wnt signaling pathway. I'm not sure what can be concluded from these data and how they can be related to regeneration. In particular, the sentence « These results are consistent with the conclusion that cell-cell communication and biosynthesis actively take place during regeneration to induce dedifferentiation/neoblast state, to regulate the proliferation of pluripotent cells and to specify the fates of the resulting cells to reconstruct the missing organs. » seems to me senseless. The final sentence « Collectively, our analyses suggested that the evolution of regeneration in earthworms might have been enhanced through the specific expansion of key genes or pathways that regulate the wound healing process or cellular proliferation. » is inappropriate, because this is not supported by the data. Even the title of the section is misleading because I don't see clearly what are these « Gene Families Related to Regeneration ». EGFR, TCAF, ZNFX, and Collagen are likely to have many roles during development and life of the animal, and it is an over-interpretation to consider that, because some of them are expressed during regeneration, their duplication might have had a role in the evolution of regeneration in *E. andrei*. The authors should completely rewrite this section, sticking to what can really be inferred by the data, or suppress this section if no clear conclusion can be drawn.

3. Third main concern is about sc-RNA-seq data. This is clearly a strong positive aspect of this paper that such an analysis has been conducted and the authors should be congratulated for that. However, the assignment of cell clusters to cell types is, to my point of view, not really convincing. In particular, I'm really not sure that expression of *sox2* is enough to demonstrate that these cells are pluripotent stem cells. In many species, including other annelids, orthologs of this gene are for example expressed in neural cells, including putative neural stem cells (which are not pluripotent) and probably also progenitors (not stem cells). Other genes whose expression is supposed to support a pluripotent stem cell fate are histone genes (H4, H14 and H2A). Their expression could maybe show that these clusters correspond to proliferating cells, but I don't see clearly how their expression can show that cells expressing these genes are pluripotent stem cells. The identification of neuron cells based on a single marker (NF70) is also not much convincing. Please note that I do not argue that cell type identification is wrong, but that it should be much more substantiated by data. My other concern is that it is a good practice to provide some experimental support of cell assignment in single cell data analyses, for example, like it is done in most or all such studies, by showing *in situ* hybridization for characteristic genes used to define identities of cell clusters. The authors should provide such data.

Other questions and suggestions :

1. As mentioned by the authors, genome sequence of the closely-related species *E. fetida* has been published. The authors could add reference to Bhambri et al. 2018 Plos One in addition to Zwarycz et al. 2015, as in fact *E. fetida* genome has been sequenced twice independently. More importantly, the authors should made some comparisons between *E. andrei* and *E. fetida* genomes. For example, one conclusion drawn from *E. fetida* genome analysis was that this species (or one of its ancestors ?) underwent extensive gene duplications. It seems to be the case in *E. andrei* as well, but did these gene duplications occurred before or after to *E.*

andrei/E. fetida divergence ? On the other way, is the LINE2 expansion described in this manuscript, specific to E. andrei or also found in E. fetida ? I found quite strange that E. fetida was not included in the diagram b of Figure 3 and in the corresponding analysis.

2. The authors chose to perform their transcriptomic analysis on anterior regeneration. I have no problem with this choice, but I think that they should briefly explain why they favored anterior over posterior regeneration (opposite choice was for example made by Bhambri et al. for E. fetida).

3. In the section « Temporal Gene Regulation Patterns in the Regeneration Response Process », the authors claimed, when discussing about the « brown module », based on their expression data and the fact that the « neoblast » term was first coined for annelid cells, that « Therefore, our analyses suggest that the brown module, including vital regulators, is initially activated and may induce the activation of pluripotent stem cells and supply necessary materials for the cell cycle. ». This is again an overstatement in particular because I think that there is no clear evidence for existence of pluripotent stem cells in their annelid model and I don't think that this can be inferred by expression of genes « involved in cellular proliferation, differentiation and programmed cell death ». Along the same line, I don't think the sentence « Therefore, our results imply that the two modules might be vital for the proliferation and maintenance of pluripotent stem cells in the regenerative processes of earthworms. » is supported by data. These overstatements should be suppressed.

4. « Convergent Genes in Earthworm and Planarian Regeneration » is a very bad title for the corresponding section. First because I don't understand what means « convergent genes ». Second, while I guess that the authors meant « convergent expression », convergence is an evolutionary hypothesis that requires some support to be proposed. Here I don't see what are arguments that would favour convergence over homology. It is possible that the three studied genes/gene families could be ancestrally involved in regeneration in bilaterians or even animals. What's also a bit annoying is this section, as in other sections in fact, is that the authors seemed to want, whatever the data they have, to find parallels and similarities with flatworms. it should not be a aim in itself! Along the same line, I don't agree with sentence in the conclusion, " We report a convergent mechanism of earthworm and planarian regeneration, including the genes EGR, RUNT, JUN and FOS." - once more nothing to support convergence.

5. The authors should give more details about their protocol for single-cell sequencing. The sentence « Earthworm single-cell sample that had undergone regeneration for up to 72 hours was prepared, and Chromium™ Single Cell Solution was applied. » is not enough. How cells are prepared in a very important step in a sc-RNA-seq analysis and the authors should provide details about how samples were treated, how cell dissociation was performed, how many worms were used, how cell sorting was done (if it was done), This is important to judge quality of the data, which is are strongly dependent of the used protocol. Along the same line, it should be indicated for the bulk transcriptomic analysis how many worms were used for each biological replicates (single worms or pools of worms ?).

6. I would have much like to have a real discussion section and not simply a few lines of conclusion. I think there are many aspects of this interesting work that deserves careful discussion.

Reviewer #2:

Remarks to the Author:

The manuscript describes the sequencing of the earthworm genome *Eisenia andrei*. Formally, a couple of assemblies have been produced before for the related *Eisenia fetida* species but at much lower quality. The authors use the genome sequence to explore the genetic pathways related to regeneration using evolutionary data and gene expression data, including at the single-cell level. The findings are fairly descriptive in nature, no mechanism is truly uncovered, but the paper makes interesting observations, especially with respect to gene clusters either co-expressed during regeneration or with respect to cell types enriched in the regeneration process. I am no expert in regeneration biology, so I cannot really estimate how much of an advance this represents in the field, and how well the results are discussed with respect to this state-of-the art, but I thought that these analyses (Figures 5 & 6) were well conducted. I have some reservations with other results, namely those presented in Figures 3 & 4, as well as some minor comments.

1. Figure 3 presents evidence for the association of LINE2 expansion with regeneration-linked differentially expressed genes (DEG). I think that the specificity of this association should be much more carefully presented. First, both the text and Fig 3e present the proportion/frequency of LINE2 elements in DEG. The two proportions are ~ 0.84 and ~ 0.75 for DEG and non-DEG respectively, two mutually exclusive classes that together represent 100% of the genes in the earthworm. So taken literally, is the conclusion that 84% of LINE2 are present in DEG and 75% in non-DEG? If so, why is the sum more than 100% of LINE2 elements? Do the authors mean that 84% and 75% of the respective gene classes contain LINE2 elements? Second, the authors should explicit in the Methods section how the LINE2 content of DEG and non-DE genes was computed. Third, figure 3f is not clear. How were these 19 LINE2/gene combinations selected out of the $\sim 6,000$ DE genes? There are described as "representative" but of what? Is the pattern proposed by the authors in any way different from what would be expected under some null hypothesis? Line 197-199, "Most neighbouring genes" relates to "most of the 19 genes presented in panel 3F" or to "most DEG genes"? If the former, then why would this be convincing since we do not know how these 19 genes were picked? If the latter, please provide exact numbers out of the ~ 6000 DE genes and a test of significance. When several LINE2 elements lie within 5 kb of a gene, do they all show a consistent expression change? Overall, I find that these results, because they are drawn from a large dataset, will always yield interesting subsets that fit whichever biological process is of interest. The authors should provide stronger evidence in favour of the LINE2 link.

2. Lines 234-261: in this section, an attempt is made at linking gene family expansion with differential gene expression. However it is not clear to me how consistent and significant these results are, compared to some simple observational results. For example, Figure 4b is used to say that "These genes may be especially important as regulatory genes during the regenerative process". The alternative is that gene families expand and contract under some other influences (unrelated to regeneration). The overlap between these expanded genes

families and the DEG gene set then captures a distribution of the former, with, as in all distribution, some samples at the extremes of this distribution (like the ZNFX1 gene family). But the manuscript does not provide any evidence that it is specifically linked to regeneration. The rest of figure 4 runs through hand-picked gene families and the related text discusses them as suggestive evidence that regeneration in earthworms evolved under the influence of (i.e. was "enhanced", line 259) through the "specific" expansion of key genes or pathways. But again, the data currently does not show this to be a specific link (a randomization test might be helpful in this regard).

Minor comments:

3. Abstract line 43: "Temporal gene trajectories" should be "Temporal gene expression trajectories ».

4. Figure 2a "Regeneration segment" should be "Regenerated segment"?

5. Supp Figure 5: I do not understand the data for time-point 0 hr: if this is the control stage against which all other time points are measured, how did the authors generate DEG genes against the control itself? The legend of the color gradient is incomplete: what does the scale bar measure?

6. Supp Table 8: what is the ranking scheme and why are some terms highlighted in red (the logic is not obvious).

7. Lines 213-214. Could the authors please indicate what threshold was used to identify the "substantially expanded" gene families?

8. Lines 312-320. I do not understand the connection between the regenerative blastema (which is formed after >96 hrs following amputation, well after the time line studied in the manuscript) and the data presented. In particular, I do not understand how it can help the authors make the conclusion line 321: "Thus, our analysis..."

Responses to reviewers' comments

Reviewer #1 (Remarks to the Author):

In this submitted manuscript, Shao and co-workers present an impressive genomic and transcriptomic study of an annelid species, the earthworm *Eisenia andrei*. Earthworms, which are key species of the soil ecosystem, display several interesting biological properties, including important regenerative abilities. These worms can indeed regenerate lost body structures, in their posterior body region, as well as in several cases (such as *E. andrei*) in their anterior region.

Earthworms, and more generally annelids, are interesting models to study regeneration, notably because they have a much more elaborated body plan (with complex organs and organ systems) than other highly studied non-vertebrate species with high regenerative abilities such as flatworms and cnidarians. A better understanding of annelid regeneration is therefore of high interest for the whole regeneration field. In addition, despite the importance of annelids, there only were until now three published full genome sequences from this group.

Shao et al. combined PacBio long-reads, Illumina short-reads and Hi-C sequencing to generate a high-quality genome assembly of *E. andrei* genome, producing what is, to my knowledge, the first chromosome level assembly of an annelid genome. The authors also performed a bulk-RNA-seq analysis at different time points of *E. andrei* anterior regeneration, identifying a large number of differentially-expressed genes during regeneration. They found that LINE2 transposable elements, which underwent a quite recent expansion in *E. andrei*, are often transcriptionally active during regeneration and might have impact on expression of adjacent genes, an interesting hypothesis for which the authors unfortunately did not show experimental evidence. The authors also studied the evolution of some gene families in *E. fetida*, and provide examples of expansion of some families by gene duplications (EGFR and TCAF families). Finally, the authors performed sc-RNA-seq at one time point after amputation, an analysis from which they drew the conclusion that major cell types of the regenerated region are pluripotent stem cells. This is clearly an important study with a lot of interesting data, and which clearly provides useful insights for our understanding of regeneration in annelids. There are however problems that should be solved by the authors.

Reply:

Thank you for your time spent on reviewing our manuscript. We sincerely appreciate your valuable comments which have definitely helped us to improve our manuscript. Please see our revisions in this manuscript version and our responses to your comments in the following.

Major concerns:

1. One of major concern is the lack of a clear description of *E. andrei* anterior regeneration.

Reply:

We thank the reviewer for this comment. According to your comment, we now describe more clearly about anterior regeneration after 1-4 body segments post-amputation. A series of morphological photos in amputation plane across regenerative stages including 0h, 6h, 12h, 24h, 48h, 72h, 4d, 5d, 6d, 7d, 14d, 18d and 28d were taken to estimate anterior regeneration compared to control *E. andrei* (Fig. 2b and Supplementary Fig 4). Meanwhile, the HE staining of transverse sections of anterior in early phases of regenerative stages was performed (Fig 2c and Supplementary Fig 5). Please check our revised main text, Fig. 2 and Supplementary Figs 4 and 5 for detail.

We also provided these descriptions here:

“Using Ki-67 immunofluorescent labeling, we found that cell proliferation initiated at 24 hours post-amputation, and at 48 and 72 hours post-amputation the proliferating cells increased rapidly and gradually migrated to the center of cross sections (Fig. 2d and Supplementary Fig 6). At 5 days post-amputation, the wound healing was fully accomplished and a small blastema (de-differentiated cells) appeared in center of the amputation plane (Supplementary Fig 4). At 6 and 7 days post-amputation, the blastema persistently experienced growth and elongation (Supplementary Fig 4). Although the newly produced body segments were not observed at 14 days post-amputation, the base of outgrowth has accumulated pigments (Supplementary Fig 4). At 18 days post-amputation, new body segments arise, and at 28 days post-amputation the obvious body segments take shape in regenerative appendages (Supplementary Fig 4).”

We provide revised Fig.2a-2d here:

Fig. 2 | Phenotypic and transcriptomic analyses during regeneration. **a**, A cartoon of time-dependent amputation and regeneration transcriptome sequencing in earthworm. **b**, Snapshots of cross sections at six time points after post-amputation. **c**, HE staining of cross sections at six time points post-amputation. The different structure layers were labeled. ED: epidermis; CM: circular muscle; LM: longitudinal muscle; IN: intestine. **d**, Detection of cell proliferation at different time points post-amputation using a marker Ki-67 by Immunofluorescent double staining. The red fluorescence represented signals and the blue fluorescence represented cell nucleus. **e**, Number comparisons of DEGs at different regeneration time points, compared to regeneration 0 hour (controls). Time points included 0, 6, 24, 48 and 72 hours. **f**, Convergent evolution of early regeneration processes in gene expression between earthworm and planarian. The early growth response genes and transcriptional factor genes respectively were compared for two species (earthworm and planarian). The planarian gene

expression changes were obtained from a previous study.

The authors took regenerated region at different time points after amputation, but we do not have any idea of what these regions look like at these different time points.

Reply:

We thank the reviewer for pointing out the issue. According to this comment, we now performed some experiments to describe more about these regions during regeneration process. We provided some high resolution graphs for each amputation plane at different stages after amputation (0h, 6h, 12h, 24h, 48h and 72h) to show general views of these regions (Fig. 2b and Supplementary Fig 4). And we also performed a series of HE labeling of transverse sections of anterior in early regeneration phases (Fig. 2c and Supplementary Fig 5). Please see our revised manuscript, Fig. 2b and 2c, and Supplementary Figs 4 and 5.

We also provided the supplementary figure 5 here:

Supplementary Fig 5 | HE staining of cross sections at six time points post-amputation. The six time points included 0, 6, 12, 24, 48 and 72 hours post-amputation. The scale size was 200 μ m. The different structure layers were labeled. ED: epidermis; CM: circular muscle; LM: longitudinal muscle; IN: intestine.

When is wound healing completed?

Reply:

A previous histological study of the earthworm, *E. andrei*¹ uncovered that at 3-5 days post-amputation, the wound healing was completed, while at 5 days post-amputation the wound healing process was fully completed because a regeneration blastema structure starts to appear.

According to your comment, we also performed a series of experiments.

Consistent with the above reference¹, we found that at 3 days post-amputation, the wound section was covered by an intact epithelium and at 5 days post-amputation, we could observe a blastema structure and at 6 days post-amputation, the blastema was obvious (Supplementary Fig 4). Therefore, together with previous study¹, we concluded that the wound healing was completed at 5 days after post-amputation. We now provide this information in the revised manuscript.

Supplementary Fig 4 | Snapshots of cross sections at 12 regenerative time points after post-amputation. At 5 days post-amputation a small blastema was formed, and at 6 days post-amputation an obvious blastema was observed.

Are there proliferating cells at these different time points?

Reply:

We thank the reviewer for pointing out this issue.

Our further experiments showed that the signals of cell proliferation at 24 hours after anterior post-amputation started to appear. At 48 and 72 hours post-amputation, especially at 72 hours after post-amputation, the proliferating cells increased rapidly and gradually extended to the center of cross section (Supplementary Fig 6).

Supplementary Fig 6 | Cell proliferation experiments using a marker Ki-67 by Immunofluorescent double staining at 0, 6 12, 24, 48 and 72 hours post-amputation. And the red fluorescence represented signals and the blue fluorescence represented cell nucleus.

When are differentiated cells or structures, such as muscles or neural cells, observed? Is the brain fully regenerated by 72 hours post-amputation?

Reply:

Thank you so much for this comment.

Actually, until now, when these differentiated cells or structures, such as muscles or neural cells, emerge remains unclear in this earthworm. During this round of review process, we used ISH of cell markers (including *TPM* and *NF70*) to track when these cells emerge. Unfortunately, due to technological failure, we didn't get positive result. However, we think it is out of scope of this study, and will not change conclusion and result of this

study.

Future more experimental studies using more different markers are necessary to answer these questions.

I think that these are crucial information to be able to make in depth use and interpretation of the nice transcriptomic data generated by the authors. This information should be provided.

Reply:

Thanks a lot for your comments. We fully agree with your points. According to your suggestions, we did a lot of experiments above. And thus we utilized the useful information to further explain our transcriptomic results. These revisions improved our manuscript well.

2. My second concern is about the section «Evolution of Gene Families Related to Regeneration», which I found not very clear and misleading. The authors identified gene families that have been expanded in *E. andrei*, including some belonging to particular pathways such as Wnt signaling pathway. I'm not sure what can be concluded from these data and how they can be related to regeneration. In particular, the sentence « These results are consistent with the conclusion that cell-cell communication and biosynthesis actively take place during regeneration to induce dedifferentiation/neoblast state, to regulate the proliferation of pluripotent cells and to specify the fates of the resulting cells to reconstruct the missing organs. » seems to me senseless. The final sentence « Collectively, our analyses suggested that the evolution of regeneration in earthworms might have been enhanced through the specific expansion of key genes or pathways that regulate the wound healing process or cellular proliferation. » is inappropriate, because this is not supported by the data. Even the title of the section is misleading because I don't see clearly what are these « Gene Families Related to Regeneration ». *EGFR*, *TCAF*, *ZNFX*, and *Collagen* are likely to have many roles during development and life of the animal, and it is an over-interpretation to consider that, because some of them are expressed during regeneration, their duplication might have had a role in the evolution of regeneration in *E. andrei*. The authors should completely rewrite this section, sticking to what can really be inferred by the data, or suppress this section if no clear conclusion can be drawn.

Reply:

Thanks very much this comment. We agree with the reviewer that we overclaim our results. According to your comment, we rewrite this section, narrow down some claims, and removed some descriptions.

Particularly, we changed the title “Evolution of Gene Families Related to

Regeneration” of this section into “Evolution of Gene Families in the Earthworm Genome”. And we also rewrite several sentences the reviewer commented,

Please see the detail in the revised manuscript.

3. Third main concern is about sc-RNA-seq data. This is clearly a strong positive aspect of this paper that such an analysis has been conducted and the authors should be congratulated for that. However, the assignment of cell clusters to cell types is, to my point of view, not really convincing. In particular, I’m really not sure that expression of *sox2* is enough to demonstrate that these cells are pluripotent stem cells. In many species, including other annelids, orthologs of this gene are for example expressed in neural cells, including putative neural stem cells (which are not pluripotent) and probably also progenitors (not stem cells). Other genes whose expression is supposed to support a pluripotent stem cell fate are histone genes (*H4*, *H14* and *H2A*). Their expression could maybe show that these clusters correspond to proliferating cells, but I don’t see clearly how their expression can show that cells expressing these genes are pluripotent stem cells. The identification of neuron cells based on a single marker (*NF70*) is also not much convincing. Please note that I do not argue that cell type identification is wrong, but that it should be much more substantiated by data. My other concern is that it is a good practice to provide some experimental support of cell assignment in single cell data analyses, for example, like it is done in most or all such studies, by showing *in situ hybridization* for characteristic genes used to define identities of cell clusters. The authors should provide such data.

Reply:

We sincerely thank the reviewer for your careful reading and professional comments, which we believe have improved greatly our manuscript.

As we know, and papers we read, many studies have validated that "Sox2 is a well-established pluripotent transcription factor that plays an essential role in establishing and maintaining pluripotent stem cells (PSCs). Together with octamer-binding transcription factor 4 and Nanog, they co-operatively control gene expression in PSCs and maintain their pluripotency." Many studies²⁻⁴ have reported *SOX2*, a master regulator of pluripotency, as a marker in pluripotent stem cells (PSCs). We now described this more clearly and cited additional references in our revised manuscript.

However, we agree with the reviewer that "In many species, including other annelids, orthologs of this gene are for example expressed in neural cells, including putative neural stem cells (which are not pluripotent) and probably also progenitors (not stem cells)". Therefore, we search for more evidences to support the identification of PSC.

(1). Histone genes (i.e., *H4*, *H1* and *H2A*) are highly expressed in clusters

(0/1/3), although they are also highly expressed in other clusters in our data (Fig. 6b, and Supplementary Figs 26 and 28).

(2). The homolog of another gene, *ACTB*, as a marker highly expressed in neoblast of planarian (*Schmidtea mediterranea*)⁵ (in this paper's Supplementary materials), was also highly co-expressed in clusters (0/1/3) in earthworm (Fig. 6c).

(3). Highly expressed markers of clusters (0/1/3) were significantly involved in those GO biological terms related to stem cells (Fig. 6d and Supplementary Fig 29).

(4). Cluster 0/1/3 located in the root of single-cell trajectories.

Furthermore, according to your comment, we performed a series of *in situ* hybridization experiments including markers of PSCs in our data and other references⁵ to validate our identifications of PSCs (Fig. 7 and Supplementary Fig 30). We believe that clusters (0/1/3) are very likely PSCs in the earthworm at 72 hours after amputation, although we can't make an absolute conclusion.

Please see our improved manuscript, Fig. 6b-6d, Fig. 7, and Supplementary Figs 26-30.

Supplementary Fig 30 | In situ hybridization of *H2B* in cross sections at 6 time points post-amputation for the earthworm. The slice size was 10 μ m. The 6 time points post-amputation included 0, 6, 12, 24, 48 and 72 hours. The red fluorescence represented positive signals and DAPI (blue fluorescence) was used

to stain cell nucleus.

To clarify the identification of neuron cells, we further search for more markers and evidences. A series of significantly highly expressed marker genes (i.e., *NF70*, *NBAS* and *AHNAK*) suggest that the cluster7 may represent putative neural cells (Supplementary Fig 32). Especially for the gene *AHNAK*, encoding neuroblast differentiation-associated protein, could function in human neural cells^{6, 7}.

Other questions and suggestions:

1. As mentioned by the authors, genome sequence of the closely-related species *E. fetida* has been published. The authors could add reference to Bhambri et al. 2018 Plos One in addition to Zwarycz et al. 2015, as in fact *E. fetida* genome has been sequenced twice independently. More importantly, the authors should made some comparisons between *E. andrei* and *E. fetida* genomes. For example, one conclusion drawn from *E. fetida* genome analysis was that this species (or one of its ancestors?) underwent extensive gene duplications. It seems to be the case in *E. andrei* as well, but did these gene duplications occurred before or after to *E. andrei*/*E. fetida* divergence? On the other way, is the LINE2 expansion described in this manuscript, specific to *E. andrei* or also found in *E. fetida*? I found quite strange that *E. fetida* was not included in the diagram b of Figure 3 and in the corresponding analysis.

Reply:

According to your comment, we added the references, Bhambri *et al.* 2018 and Paul *et al.* 2018 in our revised manuscript.

In the revision, we performed a further analysis by including the genome of *E. fetida*. Indeed, in line with your points, we found that the genomes of both *E. fetida* and *E. andrei* potentially underwent extensive gene duplications (i.e., abundant expanded gene families in earthworm branches) (Fig. 4a and Supplementary Fig 16). And our analyses of K_s distributions suggested these gene duplications occurred before *E. andrei* and *E. fetida* diverged (Fig. 4b).

Fig. 4b | *E. andrei* and *E. fetida* paranome K_s distributions and K_s distribution of one-to-one orthologs of *E. andrei* and *E. fetida*.

Please see our revisions in the section “Evolution of Gene Families in the Earthworm Genome”.

Furthermore, the *E. fetida* genome also possessed abundant content of LINE2 (~4.1%) (Fig. 3b), although the low genome assembly quality potentially underestimated the evaluation.

2. The authors chose to perform their transcriptomic analysis on anterior regeneration. I have no problem with this choice, but I think that they should briefly explain why they favored anterior over posterior regeneration (opposite choice was for example made by Bhambri et al. for *E. fetida*).

Reply:

We now explained this in the revision:

"Some studies have documented transcriptomic and some phenotypic changes of posterior regeneration in the earthworms^{16,22,23}, but very few researches are focused on the anterior regeneration¹⁴".

3. In the section «Temporal Gene Regulation Patterns in the Regeneration Response Process», the authors claimed, when discussing about the « brown module », based on their expression data and the fact that the « neoblast » term was first coined for annelid cells, that « Therefore, our analyses suggest that the brown module, including vital regulators, is initially activated and may induce the activation of pluripotent stem cells and supply necessary materials for the cell cycle. ». This is again an overstatement in particular because I think that there is no clear evidence for existence of pluripotent stem cells in their annelid model and I don't think that this can be inferred by expression of genes «involved in cellular proliferation, differentiation and programmed cell death». Along the same line, I don't think the sentence «Therefore, our results imply that the two modules might be vital for the proliferation and maintenance of pluripotent stem cells in the regenerative processes of earthworms.» is supported by data. These overstatements should be suppressed.

Reply:

We thank this reviewer for pointing out this issue. We agree with the reviewer that we overclaim our results. According to your comment, we rewrite this section, revised some description and narrow down some claims.

4. « Convergent Genes in Earthworm and Planarian Regeneration » is a very bad title for the corresponding section. First because I don't understand what means « convergent genes ». Second, while I guess that the authors meant « convergent expression », convergence is an evolutionary hypothesis that requires some

support to be proposed. Here I don't see what are arguments that would favour convergence over homology. It is possible that the three studied genes/gene families could be ancestrally involved in regeneration in bilaterians or even animals. What's also a bit annoying is this section, as in other sections in fact, is that the authors seemed to want, whatever the data they have, to find parallels and similarities with flatworms. it should not be a aim in itself! Along the same line, I don't agree with sentence in the conclusion, "We report a convergent mechanism of earthworm and planarian regeneration, including the genes EGR, RUNT, JUN and FOS." -once more nothing to support convergence.

Reply:

We sincerely thank this reviewer for this professional comment. We agreed with the reviewer that we may misuse "convergent". We now revise this section, particularly, we removed "convergence" and "convergent", and revised the title of this section as "Parallel Transcriptional Activation of Immediate Early response Genes in Earthworm and Planarian Regeneration"

We also changed our conclusion:

"our results suggest the earthworm and planarian potentially utilize a set of similar transcriptional activated immediate early response genes to regulate early regeneration process".

Please see the detail in this section of revised manuscript.

5. The authors should give more details about their protocol for single-cell sequencing. The sentence « Earthworm single-cell sample that had undergone regeneration for up to 72 hours was prepared, and Chromium™ Single Cell Solution was applied. » is not enough. How cells are prepared in a very important step in a sc-RNA-seq analysis and the authors should provide details about how samples were treated, how cell dissociation was performed, how many worms were used, how cell sorting was done (if it was done), This is important to judge quality of the data, which is are strongly dependent of the used protocol. Along the same line, it should be indicated for the bulk transcriptomic analysis how many worms were used for each biological replicates (single worms or pools of worms ?).

Reply:

We thank the reviewer for pointing out this issue. We now provided more detailed information about our protocol for the single-cell sequencing in this revised manuscript.

Earthworm single-cell samples were prepared using the following protocol:
(1) 15 Earthworms were cleaned and soil was removed using PBS or ddH2O.
(2) We used tweezers to drag the earthworms to make its head natural extended and then quickly amputated the first four body segments (the

brain is located in body segment 3~4 of the anterior). (3) Amputated earthworms were placed into soil with fertilizer and cultivated at 25 °C until to 72 hours, and then we obtained the wound healing plane segments from 15 earthworms. (4) These wound healing segments were dissociated by adding Collagenase I (500ul 1mg/ml) and then maintained 1.5~2 hours under 37°C. (5) Cells were pelleted by centrifugation at 3000rpm in 5min; the supernatant was removed and cell pellets were washed one time using 1X PBS. We then added 200ul 0.25% TE and allowed the cells to incubate for 5~10 minutes and then neutralized using 1ml 1640/DMEM including serum. (6) Cells were again pelleted at 3000rpm for 5min, the supernatant was removed and samples were resuspended in 500ul PBS. Lastly, cell samples were passed through a cell strainer with an aperture 40 ul. (7) Cells were again pelleted at 3000rpm for 5min, supernatant was removed and samples were resuspended in 200ul PBS. (8) Thus, a mixed pool of cells (from 15 earthworms) were counted and analyzed by a Flow Cytometer.

The Earthworm Single-cell RNA-Sequencing steps as follows:

Chromium™ Single Cell Solution (the experimental protocol) included the following four steps: Cell quality control. We used Countess® II Automated Cell Counter to count cells and adjusted cell concentration to 1×10^6 /ml (ideal concentration). (2) 10X marking cDNA fragments. The gel beads including 10X barcode information was first combined with the mixtures of cells and enzymes, and then they were encased by a droplet of oil with surfactant located in a "double cross" connected microfluidic. When the oil droplets flow into the storage chamber and are collected, the gel beads are dissolved and release primer sequences allowing reverse transcription into cDNA fragments. The cDNA was amplified by PCR. (3) Constructing sequencing library. We utilized Biorupter to fragment the cDNAs into 200~300bp fragments and add sequencing adaptor P5 and primer R1 to perform PCR to obtain a DNA library. (4) Cluster and sequencing. We used Qubit to qualify the sequencing library, and a high-quality sequencing library was placed onto cBot to perform Bridging PCR amplification to regenerate clusters. We then utilized Miseq sequencer to complete the sequencing.

Please see the detail in Methods section of our revised manuscript.

Additionally, we provided more detailed information about other methods. For the transcriptome (RNA-Seq), and we described more information: each amputation time point included 5 biological replicates and the wound segment of each individual was a biological replicate. We did not use pools of multiple earthworms as a biological replicate for each amputation time point. Please check our revised Methods in our manuscript.

6. I would have much like to have a real discussion section and not simply a few lines of conclusion. I think there are many aspects of this interesting work that deserves careful discussion.

Reply:

Thanks this reviewer sincerely for your valuable comments. We fully agree with your points. Therefore, according to your suggestions, we added a discussion section in our revised manuscript. Please check our main text in Discussion section.

Reviewer #2 (Remarks to the Author):

The manuscript describes the sequencing of the earthworm genome *Eisenia andrei*. Formally, a couple of assemblies have been produced before for the related *Eisenia fetida* species but at much lower quality. The authors use the genome sequence to explore the genetic pathways related to regeneration using evolutionary data and gene expression data, including at the single-cell level. The findings are fairly descriptive in nature, no mechanism is truly uncovered, but the paper makes interesting observations, especially with respect to gene clusters either co-expressed during regeneration or with respect to cell types enriched in the regeneration process. I am no expert in regeneration biology, so I cannot really estimate how much of an advance this represents in the field, and how well the results are discussed with respect to this state-of-the art, but I thought that these analyses (Figures 5 & 6) were well conducted. I have some reservations with other results, namely those presented in Figures 3 & 4, as well as some minor comments.

Reply:

We sincerely thank you for your time spent on reviewing our manuscript. We appreciate your useful comments, which have largely helped us to improve our manuscript. Please see our revisions according to your comments in our revised manuscript, and our responses to your comments as follows.

1. Figure 3 presents evidence for the association of LINE2 expansion with regeneration-linked differentially expressed genes (DEG). I think that the specificity of this association should be much more carefully presented. First, both the text and Fig 3e present the proportion/frequency of LINE2 elements in DEG. The two proportions are ~0.84 and ~0.75 for DEG and non-DEG respectively, two mutually exclusive classes that together represent 100% of the genes in the earthworm. So taken literally, is the conclusion that 84% of LINE2 are present in DEG and 75% in non-DEG? If so, why is the sum more than 100% of LINE2 elements? Do the authors mean that 84% and 75% of the respective gene classes contain LINE2 elements?

Reply:

We thank the reviewer for pointing out this issue. We are sorry for the unclear description about the frequency of LINE2 elements in this analysis.

The values in Fig. 3e mean the proportion $([\text{Gene number of DEGs containing LINE2 elements}]/[\text{Gene number of all DEGs}])$, 84%, which was significantly higher than the proportion $([\text{Gene number of non-DEGs containing LINE2 elements}]/[\text{Gene number of all non-DEGs}])$, 75%.

In addition, we further performed a more strict screening for DEGs and

removed those DEGs harboring low expression value (0) at least at one compared condition. Thus, we re-calculated the proportions (5,065/6,048 vs. 19,421/25,769) and the difference was still statistically significant ($P=7.641E-07$, χ^2 test).

We now revised the Fig. 3e and described it clearly in the figure legends. We also described it clearly in the main text:

“We discovered that the proportion of DEGs (described above) harboring LINE2 elements, was significantly higher than that of non-DEGs (background genes) harboring LINE2 elements”.

Fig. 3 | LINE2 transposable elements are related to regeneration in earthworm. **a**, Pie of the major repeat classes in earthworm genome. LINE: long interspersed nuclear elements; SINE: short interspersed nuclear elements. **b**, Comparative analyses of LINE2 contents in the genomes across different invertebrates. **c**, Divergence time of LINE2 in the earthworm genome. Kimura nucleotide distance of masked regions against their consensus sequences are automatically estimated by RepeatMasker, and the L2 element age was calculated using a mutation rate of 2.7×10^{-9} (in *C. elegans*). **d**, Distribution trends of LINE2 in the earthworm genome. **e**, Proportion of DEGs harboring LINE2 (5,065/6,048) significantly surpassed the proportion of non-DEGs harboring LINE2 (19,421/25,769) ($P=7.641E-07$, χ^2 test) during regenerative process in earthworm. **f**, Mean expression values of 44 and 119 DEL2s in 5k flanking of coding genes during regeneration process. Expression value of each DEL2 was normalized by $\text{Log}_2(\text{expression}+1)$ for each time point after post-amputation. **g**,

Similar expression profile patterns between significantly differentially expressed LINE2 located in 5-kb flanking regions of coding genes and their corresponding neighboring genes showing significant expression changes during regeneration in earthworm. Dark orange represented higher expression levels, and dark blue represented lower expression levels. The representative genes associated with regeneration were highlighted.

Second, the authors should explicit in the Methods section how the LINE2 content of DEG and non-DE genes was computed.

Reply:

Thank you so much for your valuable suggestion. We now describe it more clearly in our Methods section.

Third, figure 3f is not clear. How were these 19 LINE2/gene combinations selected out of the ~6,000 DE genes? There are described as “representative” but of what? Is the pattern proposed by the authors in any way different from what would be expected under some null hypothesis? Line 197-199, “Most neighbouring genes” relates to “most of the 19 genes presented in panel 3F” or to “most DEG genes”? If the former, then why would this be convincing since we do not know how these 19 genes were picked? If the latter, please provide exact numbers out of the ~6000 DE genes and a test of significance. When several LINE2 elements lie within 5 kb of a gene, do they all show a consistent expression change? Overall, I find that these results, because they are drawn from a large dataset, will always yield interesting subsets that fit whichever biological process is of interest. The authors should provide stronger evidence in favour of the LINE2 link.

Reply:

We thank this reviewer for pointing out this issue. We are sorry for the unclear description. We now described Fig.3f and our methods more clearly.

We divided the gtf annotation of LINE2 located in 5k flanking of the gene locus into two gtf files including 5k 5'-flanking and 5k 3'-flanking. Then, we respectively mapped our RNA-Seq at different time points (0, 6, 12, 24, 48, and 72 hours) after post-amputation to the reference genome according to the two annotations using the bowtie2 program in tophat2⁸ software. The expression abundance of each LINE2 was quantified by the cuffquant program in the cufflinks⁹, and the cuffdiff program in cufflinks⁹ was utilized to detect differentially expressed LINE2 ($P < 0.05$) between 0 hour and other time points (6, 12, 24, 48 and 72 hours) post-amputation. Thus, we further retrieved significantly differentially expressed LINE2 elements in 5'-flanking and 3'-flanking of coding genes, respectively, and identified 44 significantly differentially expressed LINE2 elements in 5'-flanking and 119 significantly differentially expressed LINE2 elements in 3'-flanking between

0 hour and at least one regeneration time point (6, 12, 24, 48, and 72 hours) ($FDR < 0.05$).

To provide stronger evidence in favor of the LINE2 link, we do additional analyses by plotting the mean expression of differentially expressed LINE2 elements in 5k 5'-flanking and 5k 3'-flanking across six different regeneration time points (Fig. 3f and supplementary Fig 12). Interestingly, we discovered that the differentially expressed LINE2 elements in 5k 5'-flanking (44 DEL2s) displayed an increasing expression trend during regeneration process (Fig. 3f and Supplementary Fig 12, $p < 0.05$, Mann-Whitney U test). And also, the differentially expressed LINE2 elements in 5k 3'-flanking (119 DEL2s) exhibited an increasing expression trend b (Fig. 3f and Supplementary Fig 12). The pattern suggested that partial LINE2 elements may potentially participate in regeneration process.

Among the neighboring genes of these significantly differentially expressed LINE2 elements (44 + 119), 19 significantly differentially expressed LINE2 elements and their neighboring genes (belonged to DEGs) exhibited similar expression trends during regeneration process. Based on previous studies on these genes, we found that most of 19 neighboring genes are involved in regeneration biology.

Therefore, we have added a section at Methods to make our analyses clearer. Please check the revisions in the updated manuscript version.

Supplementary Fig 12 | Expression profiles of differentially expressed LINE2 elements in 5k 5'-flanking and 5k 3'-flanking of coding genes during regeneration process. DEL2 represented differentially expressed LINE2 elements.

2. Lines 234-261: in this section, an attempt is made at linking gene family expansion with differential gene expression. However it is not clear to me how consistent and significant these results are, compared to some simple observational results. For example, Figure 4b is used to say that “These genes may be especially important as regulatory genes during the regenerative process”. The alternative is that gene families expand and contract under some other influences (unrelated to regeneration). The overlap between these expanded genes families and the DEG gene set then captures a distribution of the former, with, as in all distribution, some samples at the extremes of this distribution (like the *ZNFX1* gene family). But the manuscript does not provide any evidence that it is specifically linked to regeneration. The rest of figure 4 runs through hand-picked gene families and the related text discusses them as suggestive evidence that regeneration in earthworms evolved under the influence of (i.e. was “enhanced”, line 259) through the “specific” expansion of key genes or pathways. But again, the data currently does not show this to be a specific link (a randomization test might be helpful in this regard).

Reply:

Thanks for pointing out this issue. We are sorry for these unclear descriptions, and we agree with you that we overclaimed some results. We now rewrite this section to make it more clearly, and narrow down some claims.

In addition, according to your comment, we performed a randomization test. Briefly, we randomly chose 6,048 coding genes (equal to the number of total DEGs during regeneration) from whole genome wide annotated gene set (31,817 coding genes) using a sample function in R software (<https://www.r-project.org/>). Then, we computed the statistical significances for these 35 candidate significant expanded gene families harboring higher proportions of DEGs by using X^2 test between observed values and random values. We provided p values in update Fig. 4c and Supplementary Fig 18.

Again, we provided the evidence that *ZNFX1* is specifically linked to regeneration by expression profiles during regeneration process (Supplementary Fig 19).

Additionally, we also used qPCR to validate expression trends of *EGFRs* at regeneration time points, which validated our findings (Supplementary Fig 20).

Supplementary Fig 18 | A heatmap graph of 35 gene families originated from 186 significantly expanded gene families in the earthworm branch with over 10% of their family members displaying significant expression changes during regeneration. The darker color indicates more family members. The last column in heatmap represents the overlapping number between family members and DEGs. For each candidate gene family, a random test was done between observed value of DEGs in regeneration and randomly produced value by using X^2 test.

Supplementary Fig 19 | Differentially expressed analyses of *ZNF1* copies during regenerative processes in the earthworm using different time points post-amputation. The significant levels were decided by the Cuffdiff FDR correction ($P < 0.05$). The DEGs were highlighted by red.

Supplementary Fig 20 | qPCR analyses of EGFR copies during regenerative processes in the earthworm using different time points post-amputation. Four biological replicates for each time point were used and β -actin was served as a reference gene to normalize the relative mRNA expression levels. The significant levels were decided by t-test ($P < 0.05$) between 0 hour and other time points post-amputation.

Minor comments:

3. Abstract line 43: “Temporal gene trajectories” should be “Temporal gene expression trajectories ».

Reply:

Revised.

Thanks very much for carefully reading our manuscript. We now go through our manuscript, and hope that we have corrected all errors.

4. Figure 2a “Regeneration segment” should be “Regenerated segment”?

Reply:

Revised.

5. Supp Figure 5: I do not understand the data for time-point 0 hr: if this is the control stage against which all other time points are measured, how did the

authors generate DEG genes against the control itself? The legend of the color gradient is incomplete: what does the scale bar measure?

Reply:

Thanks so much for your comment. We are sorry for this unclear description. We firstly identified the 6,048 DEGs by comparing control stage to all other time points. Then, we plotted the expression profile heatmap using all of DEGs for each stage including control stage (0 hour) to see the whole expression pattern. We now described clearly this color gradient in the legend: the red color represented higher expression level and the green color represented lower gene expression level.

Please check our updated figure legend of Supplementary Fig 8 in detail.

Supplementary Fig 8 | Heatmap of 6,048 DEGs with their changed expression profiles at least in one regeneration time point. Expression profiles of DEGs are split by regenerative time-order. 6,048 DEGs were originated from comparing control stage to all other time points. The expression profile heatmap by using all of DEGs for each stage including control stage (0 hour) was plotted to show the whole expression pattern. The red color in the legend represented higher expression level and the green color represented lower gene expression level.

6. Supp Table 8: what is the ranking scheme and why are some terms highlighted in red (the logic is not obvious).

Reply:

We thank this reviewer for pointing out this issue. We are sorry for this unclear description. We highlighted these terms using red because of their biological implications potentially contributing to development and regeneration biology. Therefore, in this revision we added a clear

description for Supplementary Table 8.

7. Lines 213-214. Could the authors please indicate what threshold was used to identify the “substantially expanded” gene families?

Reply:

According to your comment, we added the threshold to identify the “substantially expanded” gene families in our Methods (Gene Family Clusters). If the copy number (gene family) of the detected branch lineage was higher than that of its closely ancestral branch, we regarded this gene family as substantially expanded gene family in this detected branch lineage. Please check our revised manuscript.

8. Lines 312-320. I do not understand the connection between the regenerative blastema (which is formed after >96 hrs following amputation, well after the time line studied in the manuscript) and the data presented. In particular, I do not understand how it can help the authors make the conclusion line 321: “Thus, our analysis...”.

Reply:

We are sorry for this unclear description. We now revise it to make it more clearly.

“The black module contains genes that exhibit upregulation within 6 hours after amputation and then gradually increase in expression until 72 hours (Fig. 5h, $r=0.49$, $P=0006$, and Supplementary Figs 23 and 24). This module presumably has an important functional role, especially at 48 and 72 hours of the early phase of regeneration, because of its sustained and increasing activity. Gene enrichment analysis found that this module is significantly enriched in genes with functions in phosphorylation, cell surface receptor, enzyme activity and ATP binding, all of which are vital for signal transduction (Supplementary Table S18). We uncovered driver genes in the black module, such as *AGRIN*, which has a higher network connectivity (intramodule membership=0.9276) and is a component of the extracellular matrix, affecting regenerative capacity and development processes in mammals^{50,51} (Fig. 5i and Supplementary Table S19). Thus, we proposed that the black module genes, with their increasing consistent temporal regulation patterns, may play an important functional role in earthworm regeneration.”

Reference

1. Park SK, Cho S-J, Park SC. Histological observations of blastema formation during earthworm tail regeneration. *Invertebr Reprod Dev* 2013, **57**(2): 165-169.
2. Leis O, Eguiara A, Lopez-Arribillaga E, Alberdi MJ, Hernandez-Garcia S, Elorriaga K, *et al.* Sox2 expression in breast tumours and activation in breast cancer stem cells. *Oncogene* 2012,

- 31(11):** 1354-1365.
3. Huangfu D, Osafune K, Maehr R, Guo W, Eijkelenboom A, Chen S, *et al.* Induction of pluripotent stem cells from primary human fibroblasts with only *OCT4* and *SOX2*. *Nat Biotechnol* 2008, **26(11):** 1269-1275.
 4. Fong H, Hohenstein KA, Donovan PJ. Regulation of self-renewal and pluripotency by Sox2 in human embryonic stem cells. *Stem Cells* 2008, **26(8):** 1931-1938.
 5. Plass M, Solana J, Wolf FA. Cell type atlas and lineage tree of a whole complex animal by single-cell transcriptomics. *Science* 2018, **360(6391):** eaaq1723.
 6. McMillan EL, Kamps AL, Lake SS, Svendsen CN, Bhattacharyya A. Gene expression changes in the MAPK pathway in both Fragile X and Down syndrome human neural progenitor cells. *Am J Stem Cells* 2012, **1(2):** 154-162.
 7. Chan SF, Huang X, McKercher SR, Zaidi R, Okamoto SI, Nakanishi N, *et al.* Transcriptional profiling of MEF2-regulated genes in human neural progenitor cells derived from embryonic stem cells. *Genom Data* 2015, **3:** 24-27.
 8. Kim D, Pertea G, Trapnell C, Pimentel H, Kelley R, Salzberg SL. Tophat2: accurate alignment of transcriptomes in the presence of insertions, deletions and gene fusions. *Genome Biol* 2013, **14(4):** R36.
 9. Ghosh S, Chan CK. Analysis of RNA-Seq data using tophat and cufflinks. *Methods Mol Biol* 2016, **1374:** 339-361.
 10. Bassat E, Mutlak YE, Genzelinakh A, Shadrin IY, Baruch Umansky K, Yifa O, *et al.* The extracellular matrix protein agrin promotes heart regeneration in mice. *Nature* 2017, **547(7662):** 179-184.
 11. Rupp F, Payan DG, Magill-Solc C, Cowan DM, Scheller RH. Structure and expression of a rat agrin. *Neuron* 1991, **6(5):** 811-823.

Reviewers' Comments:

Reviewer #1:

Remarks to the Author:

The authors have fully and convincingly addressed all my comments and criticisms.

A few details: (i) title of figure 4 and Supp. Figure 16 should be changed according to the change in the title of corresponding section in the main text (suppress "related to regeneration"); (ii) I would suppress "convergent evolution" in the abstract as it is not clear whether it is convergence or homology; (iii) In Supp. Figure 18, please explain more clearly what you mean by the number in the last column of the heatmap - I don't understand the explanation given in the figure legend ("represents the overlapping number between family members and DEGs.")

Reviewer #2:

Remarks to the Author:

I thank the authors for performing the requested changes, I am generally happy with the new manuscript. I only have two remaining points, one minor, one less so:

L204-206: I do not understand why the presence of 6.66% of LINE 2 in 5k flanking region suggests a regulatory role? Is this proportion vastly/significantly more than expected? And if so, why is that plausibly related to regulation? I think this sentence should be rephrased because the link between regulation and L2 is premature.

L273 and Supp Figure 18: After the randomisation test, the chi2 test of significance was performed 35 times, once per gene family, warranting an adjustment for multiple testing. This correction is not reported and should be. If the Bonferroni correction is applied, I believe that only one family out of 35 would remain. If so, the authors should decide if this section of the results should remain in the manuscript.

REVIEWERS' COMMENTS:

Reviewer #1 (Remarks to the Author):

The authors have fully and convincingly addressed all my comments and criticisms.

A few details:

(i) title of figure 4 and Supp. Figure 16 should be changed according to the change in the title of corresponding section in the main text (suppress "related to regeneration");

Reply:

Revised.

We have suppressed these inconsistent descriptions in revised Fig. 4 and Supplementary Figure 16.

(ii) I would suppress "convergent evolution" in the abstract as it is not clear whether it is convergence or homology;

Reply:

According to your suggestion, we have revised the description and deleted 'convergent evolution' in this revised abstract.

(iii) In Supp. Figure 18, please explain more clearly what you mean by the number in the last column of the heatmap - I don't understand the explanation given in the figure legend ("represents the overlapping number between family members and DEGs.")

Reply:

Thanks for your comments. We have re-defined the number in the last column of the HEATMAP to suppress unclear descriptions. This column number showed the number of co-shared genes between members of an assigned gene family and all of DEGs during early stages of regeneration.

Reviewer #2 (Remarks to the Author):

I thank the authors for performing the requested changes, I am generally happy with the new manuscript. I only have two remaining points, one minor, one less so:

Reply:

We thank this reviewer for the kind comments.

L204-206: I do not understand why the presence of 6.66% of LINE 2 in 5k flanking region suggests a regulatory role? Is this proportion vastly/significantly more than expected? And if so, why is that plausibly related to regulation? I think this sentence should be rephrased because the link between regulation and L2 is premature.

Reply:

We thank this reviewer for the valuable comment. We are sorry for our unclear statement. Actually, we make such suggestion because of the last whole sentence of this sentence: “Approximately 43.54% of the LINE2 elements in the earthworm genome are located in intron regions, and 6.66% are located within the 5-kb flanking regions of genes”. It is well-known that intron regions of genes may also play potential regulatory roles interacting with other genes. Thus, more than half of LINE2 in this genome (43.54%+6.66%) are located in introns of gene regions and its 5k-flanking, which suggests that these LINE2 elements may play potential regulatory roles. Here, to make our descriptions clearer, we have rephrased for this sentence in this revised manuscript.

L273 and Supp Figure 18: After the randomisation test, the chi2 test of significance was performed 35 times, once per gene family, warranting an adjustment for multiple testing. This correction is not reported and should be. If the Bonferroni correction is applied, I believe that only one family out of 35 would remain. If so, the authors should decide if this section of the results should remain in the manuscript.

Reply:

Thank you for pointing out this issue.

We deliberated upon the matter, and discussed with other scientists about this issue.

We agree with the reviewer that multiple correction of *P* value is an important method in processing those genomics big data, especially for dealing with the normalization of hundreds of *P* values from differentially expressed genes (DEGs) or other data. Multiple testing is used to exclude the possible false positive. However, we think the false positive of the two gene families we described is very low. We explain it as follow:

Before calculating *P*-value by Chi-test, we have performed a stringent analysis, by intersecting the members of the gene family and DEGs during regeneration genes. Among the 35 gene families, five gene families show significantly higher proportion of differentially expressed genes ($P < 0.05$, χ^2 test). In the result section, we only presented these five gene families not all 35 gene families, and among these five gene families, we only described the

top two, i.e. *EGFR* and *ZNF1*, which were frequently reported to participate in regenerative processes in multiple species with a strong regeneration capacity (Fraguas, et al. 2011; Sousounis, et al. 2014). Therefore, we believed that the false positive was very lower for these two gene families, although we can't exclude absolutely the possibility of false positive of the top two gene families.

In addition, Bonferroni correction is a very conservative and stringent correction, and as you see, no gene family out of 35 would remain. Therefore, as discussed in a previous study, for a finite sample size of *P* values, the multiple correction may potentially be not suitable and researchers need to balance a study's statistical significance with the magnitude of effect (Feise 2002).

Therefore, based on our points above, we suggest that it is better to retain this section in our revised manuscript.

References

- Feise RJ 2002. Do multiple outcome measures require p-value adjustment? BMC Med Res Methodol 2: 8.
- Fraguas S, Barberan S, Cebria F 2011. EGFR signaling regulates cell proliferation, differentiation and morphogenesis during planarian regeneration and homeostasis. Dev Biol 354: 87-101.
- Sousounis K, Athipposhy AT, Voss SR, Tsonis PA 2014. Plasticity for axolotl lens regeneration is associated with age-related changes in gene expression. Regeneration 1: 47-57.